# INPUT DEPENDENT SPARSE GAUSSIAN PROCESSES

## ABSTRACT

Gaussian Processes (GPs) are Bayesian models that provide uncertainty estimates associated to the predictions made. They are also very flexible due to their non-parametric nature. Nevertheless, GPs suffer from poor scalability as the number of training instances $N$ increases since their cost is cubic in $N$. To overcome this problem, sparse GP approximations are often used, where a set of $M \ll N$ inducing points is introduced during training. The location of the inducing points is learned by considering them as parameters of an approximate posterior distribution $q$. Sparse GPs, combined with variational inference for inferring $q$, reduce the cost of GPs per iteration to $\mathcal{O}(M^3)$. Critically, the inducing points determine the flexibility of the model and they are often located in regions of the input space where the latent function changes. A limitation is, however, that for some learning tasks a large number of inducing points may be required to obtain a good prediction performance. To address this limitation, we propose here to amortize the computation of the inducing points locations, as well as the parameters of the variational posterior approximation $q$. For this, we use a neural network that receives the observed data as an input and outputs the inducing points locations and the parameters of $q$. We evaluate our method in several experiments, showing that it performs similar or better than other state-of-the-art sparse variational GP approaches. However, with our method the number of inducing points is reduced drastically due to their dependency on the input data. This makes our method scale to larger datasets and have faster training and prediction times.

## 1 INTRODUCTION

Gaussian Processes (GPs) are non-parametric models that can be used to address regression and classification machine learning problems (Rasmussen & Williams, 2006). GPs become more expressive as the number of training instances $N$ grows and, since they are Bayesian models, they provide a predictive distribution that estimates the uncertainty associated to the predictions made. This uncertainty estimation or ability to know what is not known is critical in many practical applications (Gal, 2016). Nevertheless, GPs suffer from poor scalability as their training cost is $\mathcal{O}(N^3)$ per iteration due to the need of computing the inverse of a $N \times N$ covariance matrix. Another limitation is that approximate inference is needed with non-Gaussian likelihoods (Rasmussen & Williams, 2006).

Sparse approximations can improve the cost of GPs (Rasmussen & Williams, 2006). The most popular ones introduce a set of $M \ll N$ inducing points (Snelson & Ghahramani, 2006; Titsias, 2009). The inducing points and their associated posterior values completely specify the posterior process at test points. In Snelson & Ghahramani (2006), the computational gain is obtained by assuming independence among the process values at the training points given the inducing points and their values. This can also be seen as using an approximate GP prior (Quiñonero-Candela & Rasmussen, 2005). By contrast, in Titsias (2009) the computational gain is obtained by combining variational inference (VI) with a posterior approximation $q$ that has a fixed part and a tunable part. In both methods the cost is reduced to $\mathcal{O}(NM^2)$ per iteration and the inducing points, considered as model's hyper-parameters, are learned by maximizing an estimate of the marginal likelihood.

Importantly, the VI approach of Titsias (2009) maximizes a lower bound on the log-marginal likelihood as an indirect way of minimizing the KL-divergence between an approximate posterior distribution for the process values at the inducing points and the corresponding exact posterior. The advantage is that the objective is expressed as a sum over the training instances, allowing for

mini-batch training and stochastic optimization techniques to be applied on the objective (Hensman et al., 2015b). This reduces the cost to $\mathcal{O}(M^3)$ per iteration, making GPs scalable to large datasets.

In sparse approximations one often observes in practice that after the optimization process the inducing points are located in regions of the input space in which the latent function changes (Snelson & Ghahramani, 2006; Titsias, 2009; Hensman et al., 2015a; Bauer et al., 2016). Therefore, the expressive power of the model depends on the number of inducing points $M$ and their correct location on the input space. Some problems may require a large number of inducing points, in the order of thousands, to get good prediction results (Hensman et al., 2015b; Shi et al., 2020; Tran et al., 2020). This makes training inducing point based sparse GPs difficult in those problems.

There have been some attempts to improve the training cost of sparse approximations, including using different sets of inducing points for the computation of the posterior mean and variance (Cheng & Boots, 2017). Other approaches use an orthogonal decomposition of the GP that allows to introduce an extra set of inducing points with less cost (Shi et al., 2020). Finally, other methods consider a large set of inducing points, but restrict the computations for a particular data point to the nearest neighbors to that point from the set of inducing points (Tran et al., 2020).

In this work we are inspired by Tran et al. (2020) and propose a novel method to improve the training cost of sparse GPs. Our method also tries to produce a set of inducing points (and associated variational approximation $q$) that are specific of each input data point. For that, we note that some works in the literature have observed that one can learn the mappings from inputs to proposal distributions instead of directly optimizing their parameters (Kingma & Welling, 2014; Shu et al., 2018). This approach, known as amortized variational inference, is a key contribution of variational auto-encoders (VAE) (Kingma & Welling, 2014), and has also been explored in the context of GP to solve other types of problems such as multi-class classification with input noise (Villacampa-Calvo et al., 2021). Amortized inference has also been empirically shown to lead to useful regularization properties that improve the generalization performance (Shu et al., 2018).

Specifically, here we combine sparse GPs with a neural network architecture that computes, for each potential data point, the associated inducing points to be used for prediction. We also employ a neural network to carry out amortized VI to compute the parameters of the variational distribution $q$ approximating the posterior of the process values for the outputted inducing points. While the number of parameters that need optimization may increase with the use of a neural network, this approach allows for a big reduction in the total number of inducing points without losing expressive power. In particular, it enables different sets of inducing points associated to each input location. The inducing points are simply given by a mapping from the inputs provided by a neural network. We show on several experiments that the proposed method is able to perform similar or better than standard sparse GPs and competitive methods for improving the cost of sparse GPs (Tran et al., 2020; Shi et al., 2020). However, the training and prediction times of our method are much better.

## 2 GAUSSIAN PROCESSES

A Gaussian Process (GP) is a stochastic process for which any finite set of variables has a Gaussian distribution (Rasmussen & Williams, 2006). In a learning task, we use a GP as a prior over a latent function. Then, Bayes' rule is used to get a posterior for that function given the observed data. Consider a dataset $\mathcal{D} = \{(\mathbf{x}_i, y_i)\}_{i=1}^{N}$, where each scalar $y_i$ is assumed to be obtained as $y_i = f(\mathbf{x_i}) + \epsilon_i$, with $f(\cdot)$ a latent function and $\epsilon_i$ Gaussian noise with variance $\sigma^2$, *i.e.*, $\epsilon_i \sim \mathcal{N}(0, \sigma^2)$. We specify a prior distribution for $f$ in the form of a GP, which is described by a mean function $m(\mathbf{x})$ (often set to zero) and covariance function $k(\mathbf{x}, \mathbf{x}')$ such that $k(\mathbf{x}, \mathbf{x}') = \mathbb{E}[(f(\mathbf{x}) - m(\mathbf{x}))(f(\mathbf{x}') - m(\mathbf{x}'))]$. Covariance functions typically have some parameters $\boldsymbol{\theta}$. Given $\mathcal{D}$, the predictive distribution for $f$ at a new test point $\mathbf{x}^\star$ is Gaussian with mean and variance given by

$$\mu(\mathbf{x}^\star) = \mathbf{k}(\mathbf{x}^\star)^{\mathrm{T}}(\mathbf{K} + \sigma^2\mathbf{I})^{-1}\mathbf{y}\,, \quad \sigma^2(\mathbf{x}^\star) = k(\mathbf{x}^\star, \mathbf{x}^\star) - \mathbf{k}(\mathbf{x}^\star)^{\mathrm{T}}(\mathbf{K} + \sigma^2\mathbf{I})^{-1}\mathbf{k}(\mathbf{x}^\star)\,, \quad (1)$$

where $\mu(\mathbf{x}^\star)$ and $\sigma^2(\mathbf{x}^\star)$ are the prediction mean and variance, respectively. $\mathbf{k}(\mathbf{x}^\star)$ is a vector with the covariances between $f(\mathbf{x}^\star)$ and each $f(\mathbf{x}_i)$. Similarly, $\mathbf{K}$ has the covariances between $f(\mathbf{x}_i)$ and $f(\mathbf{x}_j)$ for $i, j = 1, \ldots, N$. Finally, $\mathbf{I}$ stands for the identity matrix. Popular covariances functions $k(\cdot, \cdot)$ are the squared exponential and the Matérn (Rasmussen & Williams, 2006). Their parameters, $\boldsymbol{\theta}$, and $\sigma^2$ can be found by maximizing $p(\mathbf{y})$ (Rasmussen & Williams, 2006). The computational

complexity of this approach is $O(N^3)$ since it needs the inversion of $\mathbf{K}$, a $N \times N$ matrix. This makes GPs unsuitable for large data sets.

## 2.1 SPARSE VARIATIONAL GAUSSIAN PROCESSES

Sparse approximations improve the cost of GPs. The most popular methods introduce, in the same input space as the original data, a new set of $M \ll N$ points , called the inducing points, denoted by $\mathbf{Z} = (\mathbf{z}_1, \ldots, \mathbf{z}_M)^{\mathrm{T}}$ (Snelson & Ghahramani, 2006; Titsias, 2009). Let the corresponding latent function values be $\mathbf{u} = (f(\mathbf{z}_1), \ldots, f(\mathbf{z}_M))^{\mathrm{T}}$. The inducing points are not restricted to be part of the observed data and their location can be learned during training. A GP prior is placed on $\mathbf{u}$. Namely, $p(\mathbf{u}) \sim \mathcal{N}(\mathbf{0}, \mathbf{K}_{\mathbf{Z}})$, where $\mathbf{K}_{\mathbf{Z}}$ is a matrix with the covariances associated to each pair of points from $\mathbf{Z}$. The idea is that the posterior for $f$ can be approximated in terms of the posterior for $\mathbf{u}$.

In this work we focus on a widely used variational inference (VI) approach to approximate the posterior for $f$ (Titsias, 2009). Let $\mathbf{f} = (f(\mathbf{x}_1), \ldots, f(\mathbf{x}_N))^{\mathrm{T}}$. In VI, the goal is to find an approximate posterior for $\mathbf{f}$ and $\mathbf{u}$, $q(\mathbf{f}, \mathbf{u})$, that resembles as much as possible the true posterior $p(\mathbf{f}, \mathbf{u}|\mathbf{y})$. Critically, $q$ is constrained to be $q(\mathbf{f}, \mathbf{u}) = p(\mathbf{f}|\mathbf{u})q(\mathbf{u})$, with $p(\mathbf{f}|\mathbf{u})$ fixed and $q(\mathbf{u})$ a tunable multivariate Gaussian. To find $q(\mathbf{u})$ a lower bound of the marginal likelihood is maximized. The evidence lower bound (or ELBO) is obtained via Jensen's inequality, leading to (after some simplifications):

$$\mathcal{L} = \sum_{i=1}^N \mathbb{E}_{q(\mathbf{f})}[\log p(y_i|f_i)] - \mathrm{KL}[q(\mathbf{u})|p(\mathbf{u})], \tag{2}$$

where $p(y_i|f_i)$ is the model's likelihood for the $i$-th point and $\mathrm{KL}[\cdot|\cdot]$ is the Kullback-Leibler divergence between probability distributions. In Titsias (2009), they optimize $q(\mathbf{u})$ in closed-form. The resulting expression is then maximized to estimate $\mathbf{Z}$, $\boldsymbol{\theta}$ and $\sigma^2$. This leads to a complexity of $\mathcal{O}(NM^2)$. However, if the variational posterior $q(\mathbf{u})$ is optimized alongside with $\mathbf{Z}$, $\boldsymbol{\theta}$ and $\sigma^2$, as proposed in Hensman et al. (2013), the ELBO can be expressed as a sum over training instances, which allows for mini-batch training and stochastic optimization techniques. Using stochastic variational inference (SVI) reduces the training cost to $\mathcal{O}(M^3)$ per iteration (Hensman et al., 2013). Importantly, the first term in (2) is an expectation that has closed-form solution in the case of Gaussian likelihoods. It needs to be approximated for other cases, *e.g.*, binary classification, either by quadrature or MCMC methods (Hensman et al., 2015b). The second term is the KL-divergence between the variational posterior and the prior, which can be computed analytically since they are both Gaussians.

The expressive power of the sparse GP critically depends on the number of inducing points $M$ and on their correct placement in the input space via optimizing (2) (Titsias, 2009; Hensman et al., 2015a; Bauer et al., 2016). In some problems several thousands of inducing points may be required to get good results (Hensman et al., 2015b; Shi et al., 2020; Tran et al., 2020). This makes difficult and expensive using sparse GPs in those problems. In the next section we describe how to alleviate this.

## 3 INPUT DEPENDENT SPARSE GPS

We develop a new formulation of sparse GPs which for every given input computes the corresponding inducing points to be used for prediction, and also the associated parameters of the approximate distribution $q$. To achieve this, we consider a meta-point $\tilde{\mathbf{x}}$ that is used to determine the inducing points $\mathbf{Z}$ and the corresponding $\mathbf{u}$. Namely, now $\mathbf{u}$ depends on $\tilde{\mathbf{x}}$, *i.e.*, $\mathbf{u} \sim p(\mathbf{u}|\tilde{\mathbf{x}})$. In particular, we set $p(\mathbf{u}|\tilde{\mathbf{x}}) = \mathcal{N}(\mathbf{0}, \mathbf{K}_{\mathbf{Z}(\tilde{\mathbf{x}})})$ where the inducing points $\mathbf{Z}$ depend non-linearly, *e.g.*, via a deep neural network, on $\tilde{\mathbf{x}}$. The joint distribution of $\mathbf{u}$ and $\tilde{\mathbf{x}}$ is then given by $p(\mathbf{u}, \tilde{\mathbf{x}}) = p(\mathbf{u}|\tilde{\mathbf{x}})p(\tilde{\mathbf{x}})$ for some prior distribution $p(\tilde{\mathbf{x}})$. Following (Tran et al., 2020), we can consider an implicit distribution $p(\tilde{\mathbf{x}})$. That is, its analytical form is unknown, but we can draw samples from it. Later on, we specify $p(\tilde{\mathbf{x}})$.

Note that the marginalized prior $p(\mathbf{u})$ is no longer Gaussian. However, we can show that this formulation does not impact on the prior over $\mathbf{f}$. For an arbitrary selected meta-point $\tilde{\mathbf{x}}$ we have that

$$p(\mathbf{f}, \mathbf{u}|\tilde{\mathbf{x}}) = \mathcal{N}\left(\mathbf{0}, \begin{bmatrix} \mathbf{K} & \mathbf{K}_{\mathbf{X}, \mathbf{Z}(\tilde{\mathbf{x}})} \\ \mathbf{K}_{\mathbf{Z}(\tilde{\mathbf{x}}), \mathbf{X}} & \mathbf{K}_{\mathbf{Z}(\tilde{\mathbf{x}})} \end{bmatrix}\right), \tag{3}$$

where $\mathbf{K}_{\mathbf{X}, \mathbf{Z}(\tilde{\mathbf{x}})}$ are the cross-covariances between $\mathbf{f}$ and $\mathbf{u}$. Therefore, if $\mathbf{u}$ is marginalized out in (3), the prior for $\mathbf{f}$ is the standard GP prior and does not depend on $\tilde{\mathbf{x}}$. Hence, $p(\mathbf{f}|\tilde{\mathbf{x}}) = p(\mathbf{f})$. Thus, $p(\mathbf{f}, \mathbf{u}) = \int p(\mathbf{f}, \mathbf{u}|\tilde{\mathbf{x}})p(\tilde{\mathbf{x}})d\tilde{\mathbf{x}}$ is a mixture of Gaussian densities, where the marginal over $\mathbf{f}$ is the same for every component of the mixture. In the standard sparse GP, the inducing points also have an

impact on the variational approximation $q$ via the fixed conditional distribution $p(\mathbf{f}|\mathbf{u})$ (Titsias, 2009). Therefore, we also incorporate the input dependence on $\tilde{\mathbf{x}}$ in $q$. This is done in the next section.

### 3.1 LOWER BOUND ON THE LOG-MARGINAL LIKELIHOOD

We follow Tran et al. (2020) to derive a lower bound on the log-marginal likelihood of the extended model described above. Consider a posterior approximation of the form $q(\mathbf{f}, \mathbf{u}, \tilde{\mathbf{x}}) = p(\mathbf{f}|\mathbf{u})q(\mathbf{u}|\tilde{\mathbf{x}})p(\tilde{\mathbf{x}})$, where only $q(\mathbf{u}|\tilde{\mathbf{x}})$ can be adjusted and the other factors are fixed. Using this posterior's factorization and Jensen's inequality we obtain the lower bound after some simplifications:

$$\mathcal{L} = \sum_{i=1}^{N} \int p(\tilde{\mathbf{x}}) \left[ p(f_i|\mathbf{u})q(\mathbf{u}|\tilde{\mathbf{x}}) \log p(y_i|f_i)d\mathbf{f}d\mathbf{u} - \tfrac{1}{N}\text{KL}[q(\mathbf{u}|\tilde{\mathbf{x}})|p(\mathbf{u}|\tilde{\mathbf{x}})]\right] d\tilde{\mathbf{x}}. \quad (4)$$

Now, assuming that $p(\tilde{\mathbf{x}})$ is an implicit distribution, we can draw samples from it and approximate the expectation w.r.t $p(\tilde{\mathbf{x}})$. Thus, for a meta-point sample $\tilde{\mathbf{x}}_s$ from $p(\tilde{\mathbf{x}})$, (4) is approximated as

$$\mathcal{L} \approx \sum_{i=1}^{N} \left[ \mathbb{E}_{p(f_i|\mathbf{u})q(\mathbf{u}|\tilde{\mathbf{x}}_s)}[\log p(y_i|f_i)] - \tfrac{1}{N}\text{KL}[q(\mathbf{u}|\tilde{\mathbf{x}}_s)|p(\mathbf{u}|\tilde{\mathbf{x}}_s)]\right]. \quad (5)$$

We can evaluate (5) and its gradients to maximize the original objective in (4) using stochastic optimization techniques. This is valid for any implicit distribution $p(\tilde{\mathbf{x}})$. Consider now that we use mini-batch-based training for optimization, and we set $\tilde{\mathbf{x}}_s = \mathbf{x}_i$. In this case, the value of $\tilde{\mathbf{x}}$ remains random, as it depends on the points $(\mathbf{x}_i, y_i)$ that are selected in the random mini-batch. This results in a method that computes different inducing points for each input location. In practice, we use the same sample to approximate the expectation w.r.t. $p(\tilde{\mathbf{x}})$ and the sum across the data in (5). This could introduce a bias in the objective. However, such a reusing of the samples is done in Tran et al. (2020) with good empirical results. Moreover, our experiments in Section 5 also validate this approximation.

### 3.2 AMORTIZED VARIATIONAL INFERENCE AND DEEP NEURAL NETWORKS

Maximizing the lower bound finds the optimal approximate distribution $q$. A problem, however, is that we have a potential large number of parameters to fix, corresponding to each $q(\mathbf{u}|\mathbf{x}_i)$. In particular, if we set $q(\mathbf{u}|\mathbf{x}_i)$ to be Gaussian, we will have to infer different means and covariance matrices for each different $\mathbf{x}_i$. This is expected to be memory inefficient and to make optimization more difficult. To reduce the number of parameters of our method we use amortized variational inference and specify a function that can generate these parameters for each $\mathbf{x}_i$ (Shu et al., 2018). More precisely, we set the mean and covariance matrix of $q(\mathbf{u}|\mathbf{x}_i)$ to be $\mathbf{m}(\mathbf{x}_i)$ and $\mathbf{S}(\mathbf{x}_i)$, for some non-linear functions.

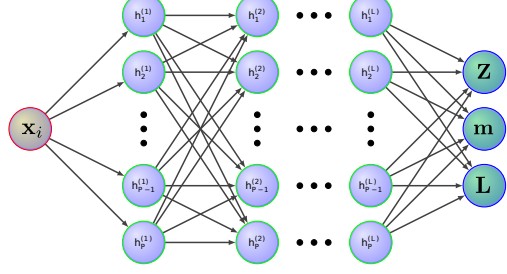

Figure 1: The network's inputs is $\tilde{\mathbf{x}}$. The outputs are the inducing points $\mathbf{Z}$, the mean vector $\mathbf{m}$ and the Cholesky factor, $\mathbf{L}$, of $q(\mathbf{u}|\mathbf{x}_i)$.

Deep neural networks (DNN) are flexible models that can describe complicated functions. In these models, the inputs go through several layers of non-linear transformations. We use these models to compute the non-linearities that generate from $\mathbf{x}_i$ the inducing points, $\mathbf{Z}(\mathbf{x}_i)$, and the means and covariances of $q(\mathbf{u}|\mathbf{x}_i)$, *i.e.*, $\mathbf{m}(\mathbf{x}_i)$ and $\mathbf{S}(\mathbf{x}_i)$. The architecture employed is displayed in Figure 1. At the output of the DNN we obtain $\mathbf{Z}$, a mean vector $\mathbf{m}$ and the Cholesky factor of the covariance matrix $\mathbf{S} = \mathbf{L}\mathbf{L}^{\text{T}}$. The maximization of the lower bound in (4) when using DNNs for the non-linearities is shown in Algorithm 1. The required expectations are computed in closed-form, in regression. In binary classification, we use 1-dimensional quadrature, as in Hensman et al. (2015a).

### 3.3 PREDICTIONS AND TRAINING COST

At test the instances are not randomly chosen. In that case, we simply set $p(\tilde{\mathbf{x}})$ to be a deterministic distribution placed on the candidate point $\mathbf{x}^\star$. The DNN is used to obtain the associated information. Namely, $\mathbf{Z}$, and the parameters of $q(\mathbf{u}|\mathbf{x}^\star)$, $\mathbf{m}$ and $\mathbf{S}$. The predictive distribution for $f(\mathbf{x}^\star)$ is:

$$f(\mathbf{x}^\star) \sim \mathcal{N}\left(\mathbf{K}_{\mathbf{x}^\star,\mathbf{Z}}\mathbf{K}_{\mathbf{Z}}^{-1}\mathbf{m}, k(\mathbf{x}^\star, \mathbf{x}^\star) + \mathbf{K}_{\mathbf{x}^\star,\mathbf{Z}}\mathbf{K}_{\mathbf{Z}}^{-1}\left(\mathbf{S} - \mathbf{K}_{\mathbf{Z}}\right)\mathbf{K}_{\mathbf{Z}}^{-1}\mathbf{K}_{\mathbf{x}^\star,\mathbf{Z}}^{\text{T}}\right). \quad (6)$$

---

**Algorithm 1** Training input dependent sparse GPs

---

**Input:** $\mathcal{D}$, $M$, neural network **NNet** with $L$ hidden layers and $P$ hidden units
**Output:** Optimal parameters of the model
   initialize neural network's weights and kernel's parameters $\boldsymbol{\theta}$
   **while** stopping criteria is False **do**
       $\text{Log}_{\text{lk}} = 0$, $\text{KL}_{\text{div}} = 0$
       gather mini-batch **Mb** of size $n$ from $\mathcal{D}$
       **for** $(\mathbf{x_i}, \mathbf{y_i})$ in **Mb** **do**
          $(\mathbf{Z}_{\mathbf{x}_i}, \mathbf{m}_{\mathbf{x}_i}, \mathbf{L}_{\mathbf{x}_i}) = \textbf{NNet}(\mathbf{x_i})$; $\text{Log}_{\text{lk}}$ += $\mathbb{E}_{q(f_i, \mathbf{u})}[\log p(y_i|f_i)]$ ;$\text{KL}_{\text{div}}$ += $\text{KL}[q(\mathbf{u}|\mathbf{x}_i)|p(\mathbf{u}|\mathbf{x}_i)]$
       $\text{ELBO} \leftarrow \frac{N}{n} \times \text{Log}_{\text{lk}} - \frac{1}{n} \times \text{KL}_{\text{div}}$
       Update parameters of the model using the gradient of ELBO

---

Given this distribution for $f(\mathbf{x}^\star)$, the probability distribution for $y^\star$ can be computed in closed form in the case of regression problems and with 1-dimensional quadrature in the case of binary classification. Note that (6) is only suitable for predictions at individual test points, as in Tran et al. (2020). This can be a limitation in applications needing covariances. As a solution, one could consider for prediction the union of all input dependent inducing points for test points. This would be inconsistent with the proposed training method. Nonetheless, such approach can also be modified to consider the union of input dependent inducing points for the training points within a mini-batch, as in (Tran et al., 2020).

The cost of our method is smaller than the cost of a standard sparse GP if a smaller number of inducing points $M$ is used. The cost of a DNN with $L$ layers, $P$ hidden units, $d_i$ dimension of the input data, and output dimension $d_o$ is $\mathcal{O}(nd_iP + nP^2L + nPd_o + n(L+1))$. The cost of the sparse GPs is $\mathcal{O}(nM^3)$, with $n$ the mini-batch size. Therefore, the cost of our method per iteration is $\mathcal{O}(nd_iP + nP^2L + nPd_o + n(L+1) + nM^3)$. Since in our method the inducing points are input dependent, we expect to obtain good prediction results even for $M$ values that are fairly small.

## 4 RELATED WORK

Early works on sparse GPs simply chose a subset of the training data for inference based on an information criterion (Csató & Opper, 2002; Lawrence et al., 2003; Seeger et al., 2003; Henao & Winther, 2012). This approach is limited in practice and more advanced methods in which the inducing points need not be equal to the training points are believed to be superior. In the literature there are several works analyzing and studying sparse GP approximations based on inducing points. Some of these works include Quiñonero-Candela & Rasmussen (2005); Snelson & Ghahramani (2006); Naish-Guzman & Holden (2007); Titsias (2009); Bauer et al. (2016); Hernández-Lobato & Hernández-Lobato (2016). We focus here on a variational approach (Titsias, 2009) which allows for stochastic optimization and mini-batch training (Hensman et al., 2013; 2015a). This enables learning in very large datasets with a cost of $\mathcal{O}(M^3)$ per iteration, with $M$ the number of inducing points.

In some problems, however, several thousands of inducing points may be needed to get good prediction results (Hensman et al., 2015b; Shi et al., 2020; Tran et al., 2020). There is hence a need to improve the cost of sparse GPs, without losing expressive power. One work addressing this task is that of Cheng & Boots (2017). In that work it is proposed to decouple the process of inferring the posterior mean and variance, allowing to consider a different number of inducing points for each one. Importantly, the computation of the mean has a linear complexity, which allows to have more expressive posterior means at a lower cost. A disadvantage is that such an approach suffers from optimization difficulties. An alternative decoupled parameterization adopts an orthogonal basis in the mean Salimbeni et al. (2018a). Such a method can be considered as a specific case of Shi et al. (2020). There, the authors introduce a new interpretation of sparse variational approximations for GP using inducing points. For this, the GP is decomposed as a sum of two independent processes. This leads to tighter lower bounds on the marginal likelihood and new inference algorithms considering two different sets of inducing points. This enables using more inducing points at a linear cost.

Our work is closer to Tran et al. (2020). There, a mechanism is also described to consider input dependent inducing points in the context of sparse GP. However, the difference is significant. In particular, in Tran et al. (2020) a very large set of inducing points $M$ is considered initially. Then, for each input point, a subset of these inducing points is considered. This subset is obtained by finding

the $K \ll M$ nearest inducing points to the current data instance $\mathbf{x}_i$. This approach significantly reduces the cost of the standard sparse GP described in Titsias (2009). However, it suffers from the difficulty of having to find the $K$ nearest neighbors for each point in a mini-batch, which is very expensive. Therefore, the final cost is higher than what would be thought initially. Our method is expected to be better because of the extra flexibility by the non-linear relation between $\mathbf{x}_i$ and $\mathbf{Z}$ given by the DNN. Furthermore, the DNN is expected to make a better use of GPU acceleration.

Another method to improve the training cost of GP is described in Wilson & Nickisch (2015); Evans & Nair (2018); Gardner et al. (2018). It consists in placing the inducing points on a grid. This allows to perform fast computation exploiting the inducing points structure. One can easily consider values for $M$ that are even larger than $N$. However, to get such benefits the inducing points need to be fixed due to the structure constraints. This may be detrimental in high-dimensional problems.

Instead of using inducing points, there are some works that scale GPs by approximating the posterior GP process using an inference network (Shi et al., 2019; Sun et al., 2019). An inference network receives some random noise and outputs function values for each input. Particular examples include among others Bayesian DNNs. Inference networks are expected to lead to flexible stochastic processes. However, it is difficult to enforce that the approximate posterior process looks similar to the prior GP in regions where there is no data. For this, approximate inference is carried out on a finite subset of points chosen at random from the input space. This is expected to lead to poor results in high-dimensional spaces. Moreover, another problem of using an inference network is that tuning the prior GP hyper-parameters is challenging and has often to be done in a separate step.

Amortized variational inference (Shu et al., 2018) has also been explored in the context of GPs in Villacampa-Calvo et al. (2021). There, input noise is considered in a multi-class learning problem and the performance of the final model is improved by amortizing the variational parameters of the posterior approximation for the noiseless inputs, using a DNN that receives both $\mathbf{x}_i$ and $y_i$.

Other sparse GPs in the literature do not fully rely on inducing points, *e.g.*, (Tresp, 2000; Snelson, 2007; Gramacy & Apley, 2015). These techniques, however, cannot use, in general, stochastic optimization and do not scale to very large problems. Finally, sparse GPs, and our method, can benefit from natural-gradients (Salimbeni et al., 2018b). They could result in an orthogonal improvement.

## 5 EXPERIMENTS

We evaluate the performance of the proposed method, to which we refer to as Input Dependent Sparse GP (IDSGP). We consider both regression and binary classification with a probit likelihood. In this later case, we approximate the expectation in the lower bound using 1-dimensional quadrature, as in Hensman et al. (2015a). The code of the proposed method in Tensorflow 2.0 (Abadi et al., 2015) is provided in the supplementary material. In the experiments we compare results with the standard variational sparse GP (Titsias, 2009). We refer to such a method as VSGP. We also compare results with two of the methods described in Section 4. Namely, the sparse within sparse GP (SWSGP) described in Tran et al. (2020), and the sparse GP based on an orthogonal decomposition that allows to consider two different sets of inducing points (Shi et al., 2020). We refer to this last method as SOLVE. All methods use a Matérn $3/2$ covariance function (Rasmussen & Williams, 2006). The DNN architecture employed in IDSGP is described in detail in Appendix B.

### 5.1 TOY PROBLEMS

We show the posterior mean and standard deviation of each method on a 1-dimensional regression problem (Snelson & Ghahramani, 2006). We compare results with a full GP. Figure 2 shows the results obtained, including the learned locations of the inducing points. In the case of IDSGP we show the locations of the inducing points for the point represented with a star at $x = 3.9$. The number of inducing points, for each method, are indicated in the figure's caption. We consider a small number of inducing points to study the benefits of having input dependent inducing points. IDSGP uses smaller number of inducing points than the other methods. The figure shows that, in regions with observed data, IDSGP's predictions look closer to those of the full GP. Appendix D.1 has results for an increasing number of inducing points $M$. They show that as $M$ increases IDSGP becomes more similar to the full GP. Figure 3 shows the decision boundary of each method on the banana classification dataset (Hensman et al., 2013). IDSGP produces the most accurate boundaries.

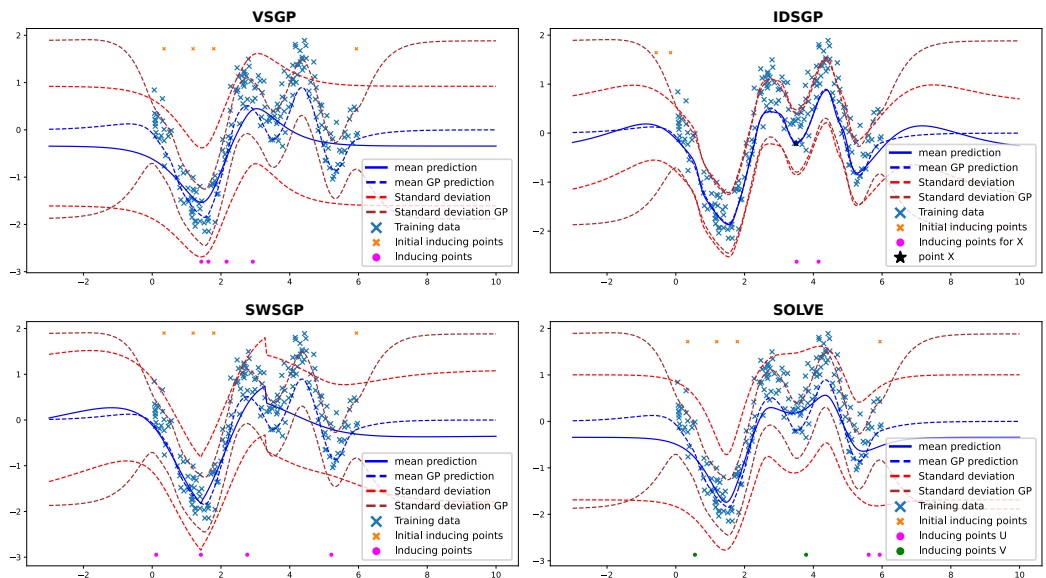

Figure 2: Toy data set with $N = 200$ points. Initial and final locations for the inducing points are shown on the top and bottom of each figure. In IDSGP, the inducing points correspond to the point drawn with star. The posterior mean and standard deviation of full GP are shown with blue and brown dashed lines, respectively. VSGP method with $M = 4$. IDSGP with $M = 2$ and a neural network with 2 layers with 50 units. SWSGP with $M = 4$ and 2 neighbors. SOLVE with $M_1 = M_2 = 2$.

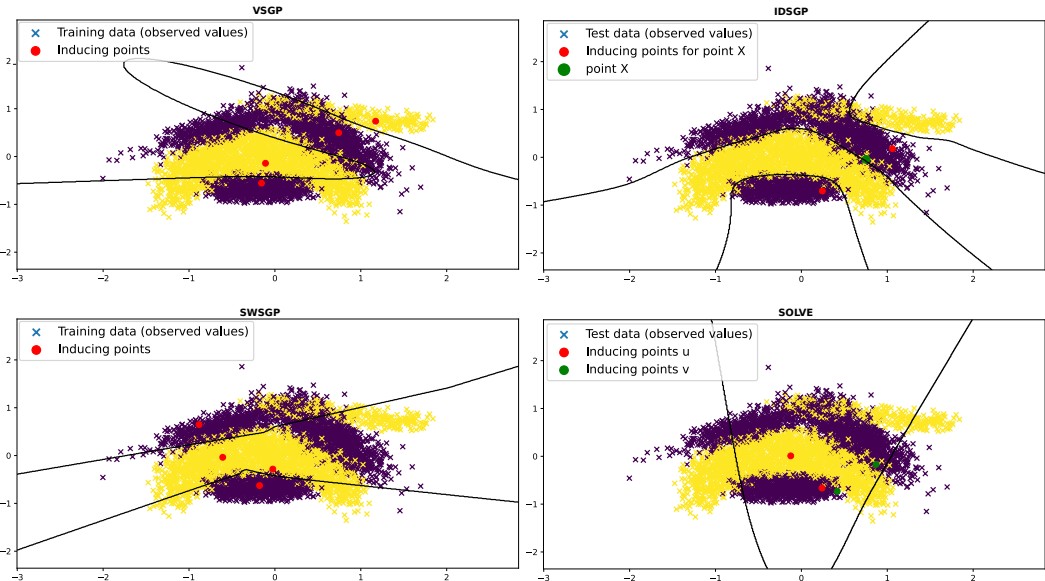

Figure 3: Banana classification data set with $N = 5300$ points. The final location of inducing points are shown inside the figures. For IDSGP, we show the location of inducing points related to the green colored point. VSGP with $M = 4$. IDSGP with $M = 2$ and a neural network with 2 hidden layers each contains 50 hidden nodes. SWSGP with $M = 4$ and 2 neighbors. SOLVE with $M_1 = M_2 = 2$.

## 5.2 EXPERIMENTS ON UCI DATASETS

We consider several regression and binary classification datasets extracted from the UCI repository (Dua & Graff, 2017). The number of inducing points of IDSGP is set to $M = 15$. In SOLVE we use $M_1 = 1024$ and $M_2 = 1024$ inducing points. In VSGP we set $M = 1024$. In SWSGP we set $M = 1024$ and $K = 50$ neighbors. All the methods are trained using ADAM (Kingma & Ba, 2015)

with a mini-batch size of 100 and a learning rate of 0.01. In the classification setting we use the same setup, but the number of inducing points of IDSGP is set to $M = 3$. All methods are trained on a Tesla P100 GPU with 16GB of memory. On each dataset we use 80% of the data for training and the rest for testing. We report results across 5 splits of the data since the datasets are already quite big.

The average negative test log-likelihood of each method on each dataset is displayed in Table 1, for the regression datasets, and in Table 2, for the classification datasets, respectively. The average rank of each method is also displayed at the last row of each table. The RMSE and prediction accuracy results are similar to those displayed here. They can be found in Appendix D.3 and D.4. Each table also shows the number of instances $N$ and dimensions $d$ of each dataset. We observe that in the regression datasets, the proposed method, IDSGP, obtains best results in 6 out of the 8 datasets. IDSGP also obtains the best average rank (closer to always performing best on each train / test data split). This is remarkable given that IDSGP a much smaller number of inducing points (*e.g.*, $M = 15$ for IDSGP vs. $M = 1024$ for VSGP). In classification, however, all the methods seem to perform similar to each other and the differences between them are smaller. Again IDSGP uses here a smaller number of $M = 3$ inducing points. Increasing $M$ in IDSGP does not improve the results.

Table 1: Avg. neg. test log-likelihood values for the UCI regression datasets. The numbers in parentheses are standard errors. Best mean values are highlighted in bold face.

|  | $N$ | $d$ | VSGP | SOLVE | SWSGP | IDSGP |
|---|---|---|---|---|---|---|
| Kin40k | 32,000 | 8 | -0.047 (0.003) | -0.415 (0.006) | -0.110 (0.007) | **-1.461 (0.019)** |
| Protein | 36,584 | 9 | 2.848 (0.002) | 2.818 (0.003) | 2.835 (0.002) | **2.775 (0.007)** |
| KeggDirected | 42,730 | 19 | -1.955 (0.013) | -1.756 (0.073) | -2.256 (0.012) | **-2.410 (0.012)** |
| KEGGU | 51,686 | 26 | -2.344 (0.012) | -2.531 (0.015) | -2.396 (0.006) | **-2.908 (0.042)** |
| 3dRoad | 347,899 | 3 | 3.691 (0.006) | 3.726 (0.010) | 3.879 (0.026) | **3.399 (0.009)** |
| Song | 412,276 | 90 | 3.613 (0.003) | **3.608 (0.002)** | 3.618 (0.004) | 3.637 (0.002) |
| Buzz | 466,600 | 77 | 6.272 (0.012) | 6.297 (0.009) | **6.137 (0.008)** | 6.317 (0.055) |
| HouseElectric | 1,639,424 | 6 | -1.737 (0.006) | -1.743 (0.005) | -1.711 (0.010) | **-1.774 (0.004)** |
| Avg. Ranks |  |  | 3.125 (0.125) | 2.475 (0.156) | 2.850 (0.150) | **1.550 (0.172)** |
| # Inducing points |  |  | 1024 | 1024 / 1024 | (K=50) / 1024 | 15 |

Table 2: Avg. test neg. log-likelihood values for the UCI classification datasets. The numbers in parentheses are standard errors. Best mean values are highlighted in bold face.

|  | $N$ | $d$ | VSGP | SOLVE | SWSGP | IDSGP |
|---|---|---|---|---|---|---|
| MagicGamma | 15,216 | 10 | **0.308 (0.004)** | 0.314 (0.005) | 0.371 (0.005) | 0.311 (0.002) |
| DefaultOrCredit | 24,000 | 30 | **0.000 (0.000)** | **0.000 (0.000)** | **0.000 (0.000)** | **0.000 (0.000)** |
| NOMAO | 27,572 | 174 | 0.113 (0.004) | **0.103 (0.004)** | 0.134 (0.004) | 0.121 (0.004) |
| BankMarketing | 36,169 | 51 | 0.206 (0.001) | **0.199 (0.001)** | 0.304 (0.021) | 0.209 (0.002) |
| Miniboone | 104,051 | 50 | 0.151 (0.001) | **0.142 (0.001)** | 0.180 (0.007) | 0.153 (0.001) |
| Skin | 196,046 | 3 | 0.005 (0.000) | 0.005 (0.000) | 0.006 (0.001) | **0.003 (0.000)** |
| Crop | 260,667 | 174 | 0.003 (0.000) | 0.003 (0.000) | **0.002 (0.000)** | 0.003 (0.000) |
| HTSensor | 743,193 | 11 | 0.003 (0.001) | **0.001 (0.000)** | 0.030 (0.009) | 0.005 (0.001) |
| Avg. Ranks |  |  | 2.425 (0.143) | **1.775 (0.158)** | 3.175 (0.182) | 2.625 (0.155) |
| # Inducing points |  |  | 1024 | 1024 / 1024 | (K=50) / 1024 | 3 |

In these experiments we also measure the average training time per epoch, for each method. The results corresponding to the UCI regression datasets are displayed in Table 3. The results for the UCI classification datasets are found in Appendix D.4. They look very similar to ones reported here. We observe that the fastest method in terms of training time is the proposed approach. Namely, IDSGP. Nevertheless, the speed-up obtained is impaired by the overhead of having to compute the output of the DNN and update its parameters. IDSGP also results in fastest prediction times than VSGP, SOLVE or SWSGP. See Appendix D.3 and D.4 for further results showing average prediction times.

Table 3: Average training time per epoch across the 5 splits for the UCI regression datasets. The numbers in parentheses are standard errors. Best mean values are highlighted.

|  | Kin40k | Protein | KeggDirected | KEGGU | 3dRoad | Song | Buzz | HouseElectric |
|---|---|---|---|---|---|---|---|---|
| VSGP | 591.7 (0.58) | 737.2 (1.16) | 932.7 (2.56) | 1128.1 (3.78) | 7880.9 (66.79) | 9777.7 (42.84) | 9901.0 (146.07) | 32784.2 (190.18) |
| SOLVE | 1739.3 (0.45) | 2015.9 (0.66) | 2357.3 (1.70) | 2909.1 (1.19) | 19567.1 (10.34) | 23196.6 (98.35) | 25769.5 (20.12) | 92214.9 (452.18) |
| SWSGP | 875.7 (0.68) | 1023.5 (0.35) | 1220.6 (1.89) | 1458.0 (5.57) | 10203.4 (12.03) | 12241.7 (62.01) | 13371.5 (12.34) | 46163.3 (427.23) |
| IDSGP | **190.3 (0.75)** | **371.5 (1.25)** | **533.0 (1.73)** | **693.7 (5.77)** | **4070.1 (201.09)** | **4296.5 (25.03)** | **3640.4 (33.36)** | **16352.2 (90.15)** |

## 5.3 LARGE SCALE DATASETS

A last set of experiments considers two very large datasets. The first dataset is the Airlines Delay binary classification dataset, as described in Hernández-Lobato & Hernández-Lobato (2016), with $N = 2,127,068$ data instances and $d = 8$ attributes. The second dataset is the Yellow taxi dataset, as described in Salimbeni & Deisenroth (2017), with $N = 1$ billion data-points and $d = 9$ attributes. In each dataset we use a test set of $10,000$ instances chosen at random. The number of inducing points is set to be equal to $M = 50$ in IDSGP. In the other methods, we use the same number of inducing points as in the previous section. The mini-batch size is set to $100$. Training is also performed on the same GPU as in the previous section. The ADAM learning rate is set to $0.001$.

The average negative test log-likelihood of each method is displayed in Figure 4, for each dataset. We report performance in terms of the training time, in a $\log_{10}$ scale. The results corresponding to the RMSE are very similar to the ones displayed here. They can be found in Appendix D.5. We observe that the proposed method IDSGP performs best on each dataset. In particular, it obtains a better performance in a smaller computational time. We believe this is a consequence of using a smaller number of inducing points, and also because of the extra flexibility that the DNN provides for specifying the locations of the inducing points.

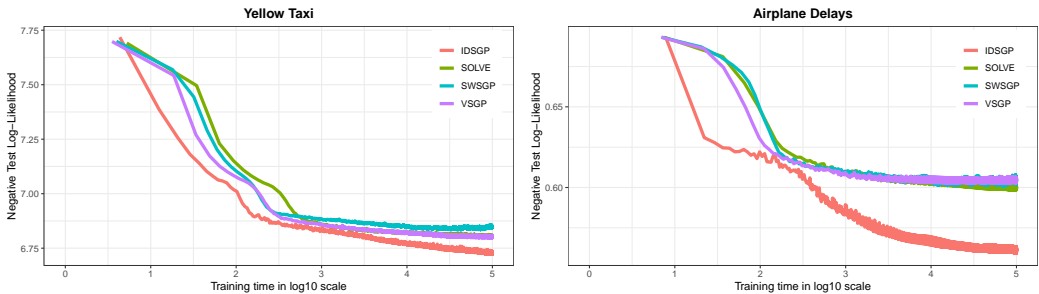

Figure 4: Negative log-likelihood on the test set for each method as a function of the training time in seconds, in $\log_{10}$ scale, for the Yellow taxi and the Airline delays datasets. Best seen in color.

## 6 CONCLUSIONS

Gaussian processes (GPs) are flexible models for regression and classification. However, they have a cost of $\mathcal{O}(N^3)$ per iteration with $N$ the number of training points. Sparse approximations based on $M \ll N$ inducing points reduce such a cost to $\mathcal{O}(M^3)$. A problem, however, is that in some situations a large number of inducing points have to be used in practice, since they determine the flexibility of the resulting approximation. There is hence a need to reduce their training cost.

We have proposed here input dependent sparse GP (IDSGP), a method that can improve the training time and the flexibility of sparse GP approximations. IDSGP uses a deep neural network (DNN) to output specific inducing points for each point at which the predictive distribution of the GP needs to be computed. The DNN also outputs the parameters of the corresponding variational approximation on the inducing values associated to the inducing points. IDSGP can be obtained under a formulation that considers an implicit distribution for the input instance to the DNN. Importantly, such a formulation is shown to keep intact the GP prior on the latent function values associated to the training points.

The extra flexibility provided by the DNN allows to significantly reduce the number $M$ of inducing points used in IDSGP. Such a model provides similar or better results than other sparse GP approximations from the literature at a smaller training cost. IDSGP has been evaluated on several regression and binary classification problems from the UCI repository. The results obtained show that it improves the quality of the predictive distribution and reduces the training cost of sparse GP approximations. Better results are most of the times obtained in regression problems. In classification problems, however, the performances obtained are similar to those of the state-of-the-art, although the training and prediction times are always shorter. The scalability of IDSGP is also illustrated on massive datasets for regression and binary classification of up to $1$ billion points. There, IDSGP also obtains better results than alternative sparse GP approximations at a smaller training cost.

ETHICS STATEMENT

The authors acknowledge to have read and commit to adhering to the ICLR Code of Ethics [1]. We do not see any direct potential negative societal impact, because this paper focuses on the development of a new methodology. We believe these would be indirect through the particular application in which the proposed method is used. As one of the main advantages of GPs is that they provide uncertainty estimates associated with the predictions made, we think the potential harm of these models in society could arise in applications when this uncertainty estimation is critical. For example, an AI system in which the decisions made can have an influence on people's life, such as autonomous vehicles or automatic medical diagnosis tools.

REPRODUCIBILITY STATEMENT

Most of the details needed to reproduce this paper's results are described in Section 5. In Appendix B we give further details about the neural network architecture and initialization. We provide the code used to run the experiments in the supplementary material, with implementation of our method and the rest of methods of the experimental comparison. Regarding the pre-processing of the data, you can find all the information needed to reproduce this step in Appendix A.

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

## A    DATASETS PRE-PROCESSING

All the datasets are publicly available. The UCI repository datasets can be downloaded from the repository (Dua & Graff, 2017). Yellow taxi dataset was preprocessed following Salimbeni & Deisenroth (2017) and downloaded from `https://www1.nyc.gov/site/tlc/about/tlc-trip-record-data.page`, where we have used data records from year 2015. Similarly, the Airlines Delay dataset was preprocessed following Hernández-Lobato & Hernández-Lobato (2016) and was downloaded from `https://community.amstat.org/jointscsg-section/dataexpo/dataexpo2009`, keeping only the records from January 2008 to April 2008. All datasets have been standardized using scikit-learn's built-int StandardScaler class (Pedregosa et al., 2011), which removes the mean and scales to unit variance.

## B    NEURAL NETWORK ARCHITECTURE

About the choice of architecture for the DNN we have tried to keep it small in order to take more advantage of the computational gain of the amortized scheme. In particular, we used a 2 hidden-layer with 50 hidden units network for the toy problems in Section 5.1, a 1 layer with 50 hidden units network for the UCI datasets in Section 5.2 and a 2 layer with 25 hidden units for the large scale datasets in Section 5.3. We used ReLu activation functions. Keeping the network small reduces the number of parameters to optimize making the optimization process easier. In all problems we are using fully-connected layers with batch normalization and no skip layers. Regarding the initialization of the weights, all were initialized using the Glorot initialization (Glorot & Bengio, 2010). In our experiments we did not exhaustively explore the DNN architecture. This choice of architecture and initialization was based on some preliminary tests done before running the experiments. This does not mean that this is the best possible configuration. We did not optimize the architecture of the neural network. In practical applications, we suggest to run some preliminary tests in order to choose a configuration that performs well. The main suggestion, however, is to keep the network small as the input dependence will make the model expressive enough to still get very good results.

## C    CHOOSING THE NUMBER OF INDUCING POINTS

We have observed that our proposed method IDSGP performs well in general with a fairly small number of inducing points, much smaller than the number of inducing points used in the other methods. Namely, SOLVE, VSGP and SWSGP. This is probably related to the extra flexibility of having input-dependent inducing points in IDSGP. In very large datasets we recommend using around $M = 50$ inducing points. In medium-size regression datasets $M = 15$ inducing points seem enough. In medium-size binary classification datasets, however, a smaller number of inducing points is enough $M = 3$. We believe the reason is that binary classification problems require less complicated latent functions. We did not optimize the number of inducing points. In practical applications, we suggest to run some preliminary tests in order to choose a number of inducing points that performs well.

## C.1 KL-Divergence Minimization

In this section we show that maximizing (4) effectively minimizes the KL-divergence between $q(\tilde{\mathbf{x}}, \mathbf{f}, \mathbf{u})$ and $p(\tilde{\mathbf{x}}, \mathbf{f}, \mathbf{u}|\mathbf{y})$. In particular,

$$
\begin{aligned}
\text{KL}(q(\tilde{\mathbf{x}}, \mathbf{f}, \mathbf{u})|p(\tilde{\mathbf{x}}, \mathbf{f}, \mathbf{u}|\mathbf{y})) &= -\int q(\tilde{\mathbf{x}}, \mathbf{f}, \mathbf{u}) \log \frac{p(\mathbf{y}, \tilde{\mathbf{x}}, \mathbf{f}, \mathbf{u})}{q(\tilde{\mathbf{x}}, \mathbf{f}, \mathbf{u})} d\mathbf{x} d\mathbf{f} d\mathbf{u} + \text{const.} \\
&= -\int q(\tilde{\mathbf{x}}, \mathbf{f}, \mathbf{u}) \log \frac{p(\mathbf{y}|\mathbf{f})p(\mathbf{f}|\mathbf{u})p(\mathbf{u}|\tilde{\mathbf{x}})p(\tilde{\mathbf{x}})}{p(\mathbf{f}|\mathbf{u})q(\mathbf{u}|\tilde{\mathbf{x}})p(\tilde{\mathbf{x}})} d\mathbf{x} d\mathbf{f} d\mathbf{u} + \text{const.} \\
&= -\int q(\tilde{\mathbf{x}}, \mathbf{f}, \mathbf{u}) \log \frac{p(\mathbf{y}|\mathbf{f})p(\mathbf{u}|\tilde{\mathbf{x}})}{q(\mathbf{u}|\tilde{\mathbf{x}})} d\mathbf{x} d\mathbf{f} d\mathbf{u} + \text{const.} \\
&= -\int q(\mathbf{f}, \tilde{\mathbf{x}}, \mathbf{u}) \log p(\mathbf{y}|\mathbf{f}) d\mathbf{f} d\mathbf{x} d\mathbf{u} \\
&\quad + \int q(\mathbf{u}, \tilde{\mathbf{x}}) \log \frac{p(\mathbf{u}|\tilde{\mathbf{x}})}{q(\mathbf{u}|\tilde{\mathbf{x}})} d\mathbf{x} d\mathbf{u} + \text{const.} \\
&= -\mathbb{E}_q[\log p(\mathbf{y}|\mathbf{f})] + \mathbb{E}_{p(\tilde{\mathbf{x}})}[\text{KL}(q(\mathbf{u}|\tilde{\mathbf{x}})|p(\mathbf{u}|\tilde{\mathbf{x}}))] \\
&= -\mathcal{L} + \text{const.} ,
\end{aligned}
\tag{7}
$$

where we have used that the posterior is equal to the joint $p(\tilde{\mathbf{x}}, \mathbf{f}, \mathbf{u}, \mathbf{y})$ divided by a normalization constant, *i.e.*, the marginal likelihood. Moreover, $\mathcal{L}$ is simply the lower bound defined in (4). Therefore, maximizing $\mathcal{L}$ effectively leads to the minimization of the KL-divergence between $q(\tilde{\mathbf{x}}, \mathbf{f}, \mathbf{u})$ and $p(\tilde{\mathbf{x}}, \mathbf{f}, \mathbf{u}|\mathbf{y})$.

# D    Extra experimental results

In this section, we include some extra results that do not fit in the main manuscript. Namely, the RMSE in the test set results and prediction times for the UCI regression datasets, and the accuracy in the test set, training and prediction times for the UCI classification datasets. In both cases, the setup is the same as described in Section 5 and the results are similar that the ones obtained in terms of the negative test log likelihood and training times in that section. Finally, we include similar plots to those in Section 5.3 but in terms of the test RMSE for the Yellow Taxi dataset and in terms of the test classification error for the Airline Delays dataset.

## D.1    Toy regression datasets

Our method looks more and more similar to the full GP as number of inducing points $M$ increases. However, with a small number of inducing points, it gives similar results to those of the full GP and similar to the results obtained when more inducing points are considered, which does not happen in the other methods. This is probably due to the extra flexibility of the neural network. The figures below (Figures 6 to 8) show the results of each method on the toy regression problem as we increase the number of inducing points $M$. For $M = 128$ IDSGP gives almost the same results as VSGP.

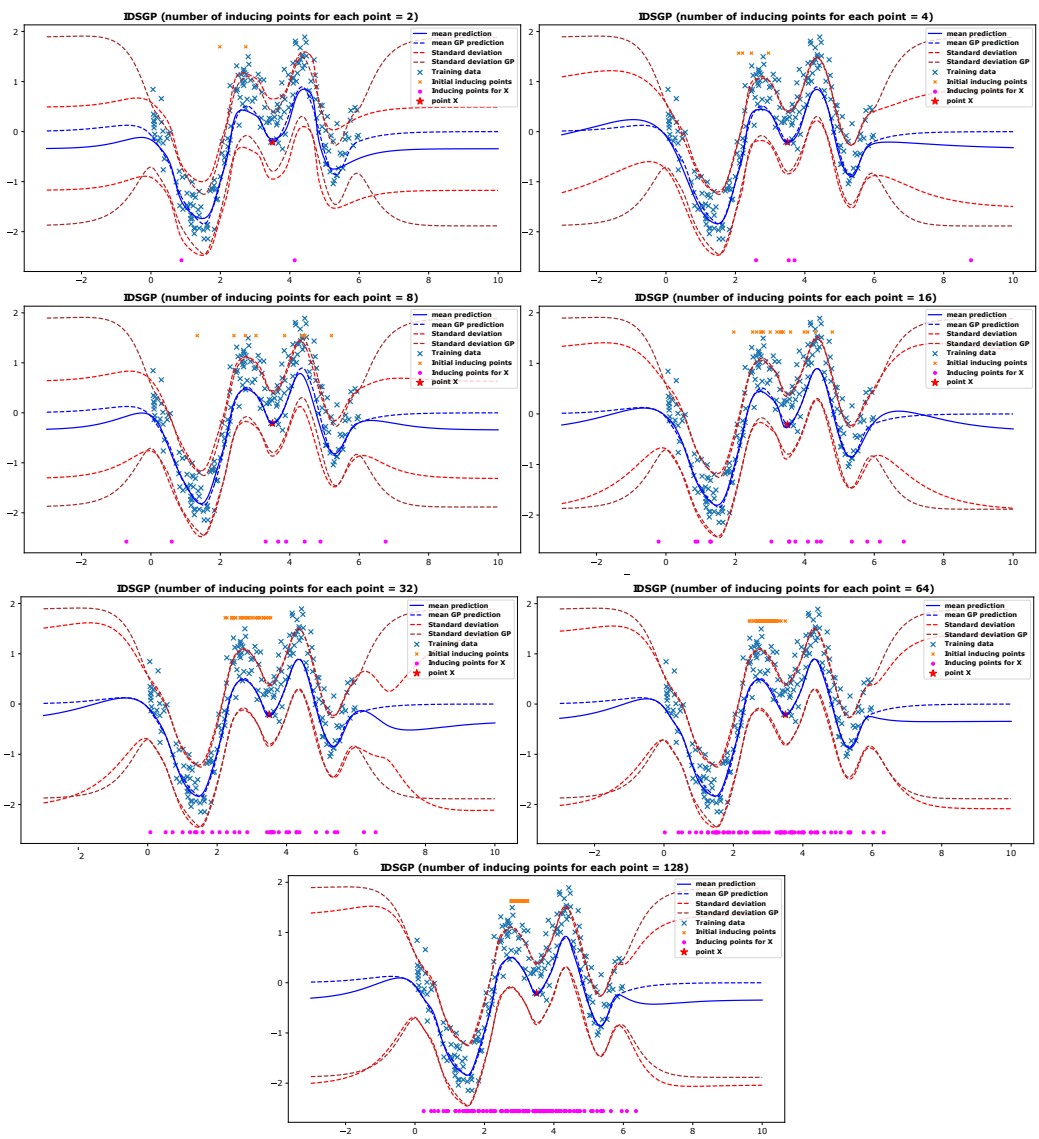

Figure 5: Toy regression example by varying number of inducing points $M_x = \{2, 4, 8, 16, 32, 64, 128\}$ with location of initial and final inducing points for an arbitrary selected point $x$ from training sets. The mean and standard deviation of full GP prediction are shown with dashed blue and brown lines, respectively. The blue lines and the dashed red lines are the mean and standard deviation of IDSGP.

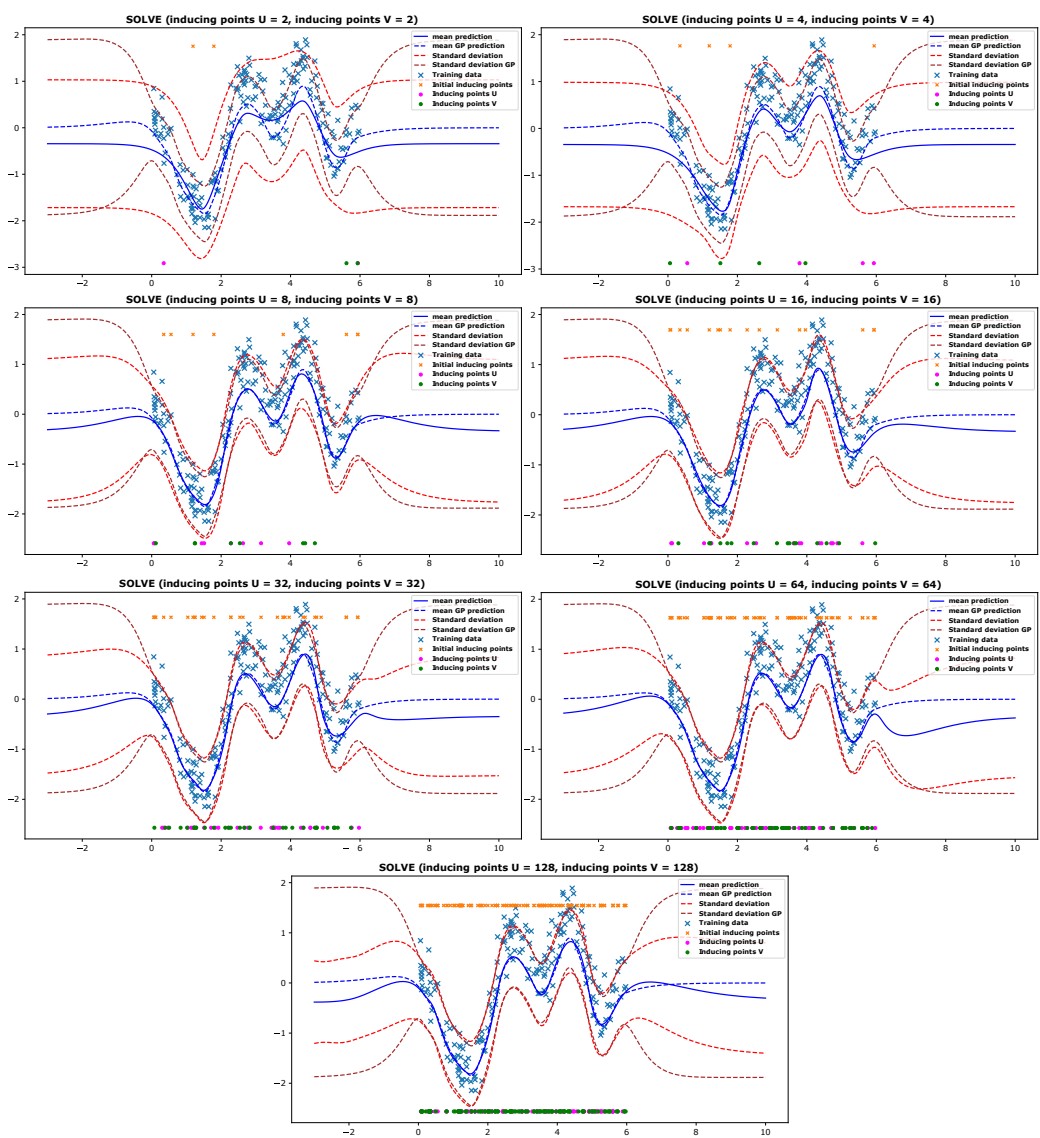

Figure 6: Toy regression example by varying number of inducing points $M_u, M_v = \{2, 4, 8, 16, 32, 64, 128\}$ with location of initial and final inducing points. The mean and standard deviation of full GP prediction are shown with dashed blue and brown lines, respectively. The blue lines and the dashed red lines are the mean and standard deviation of SOLVE.

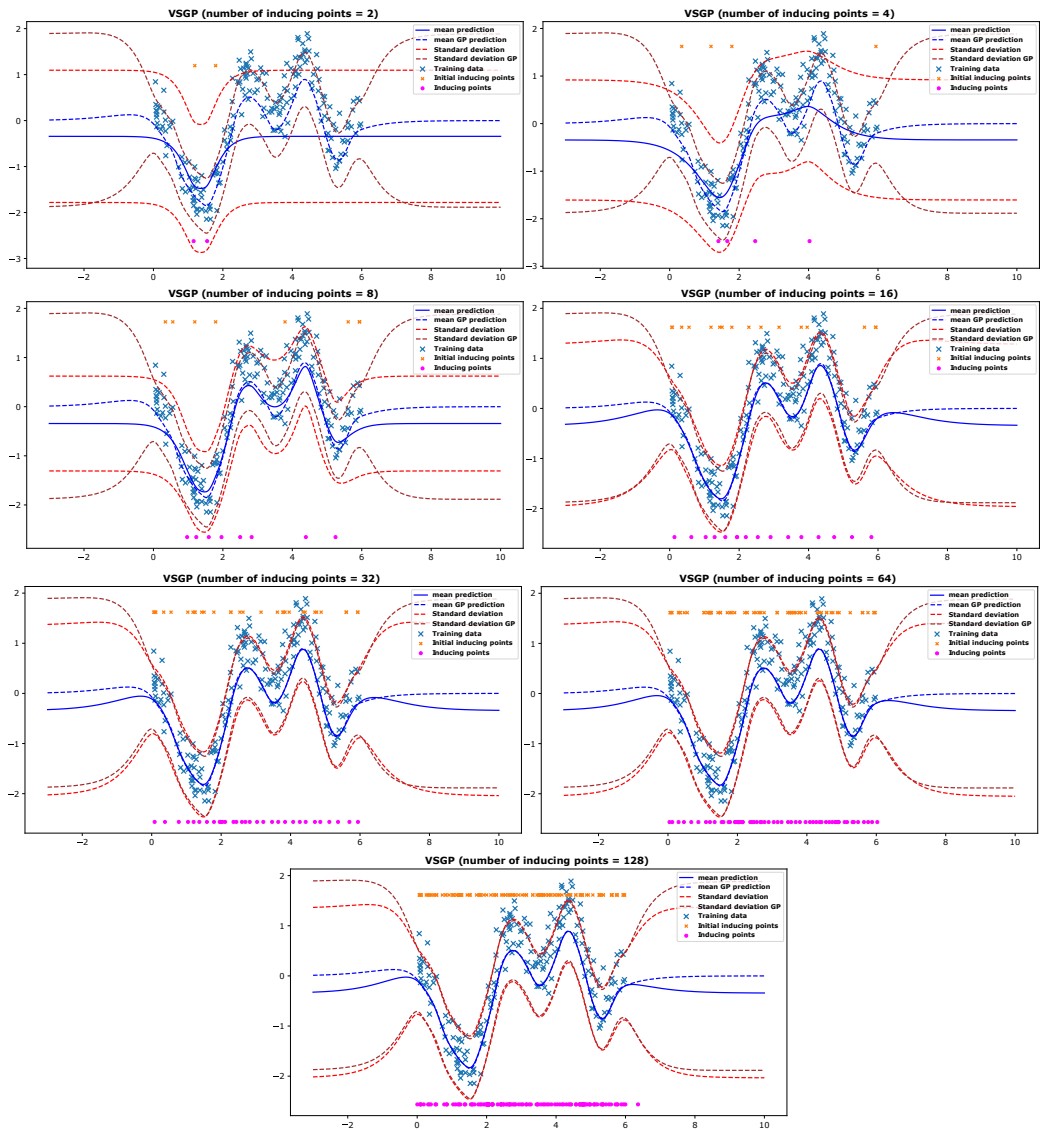

Figure 7: Toy regression example by varying number of inducing points $M = \{2, 4, 8, 16, 32, 64, 128\}$ with location of initial and final inducing points. The mean and standard deviation of full GP prediction are shown with dashed blue and brown lines, respectively. The blue lines and the dashed red lines are the mean and standard deviation of VSGP.

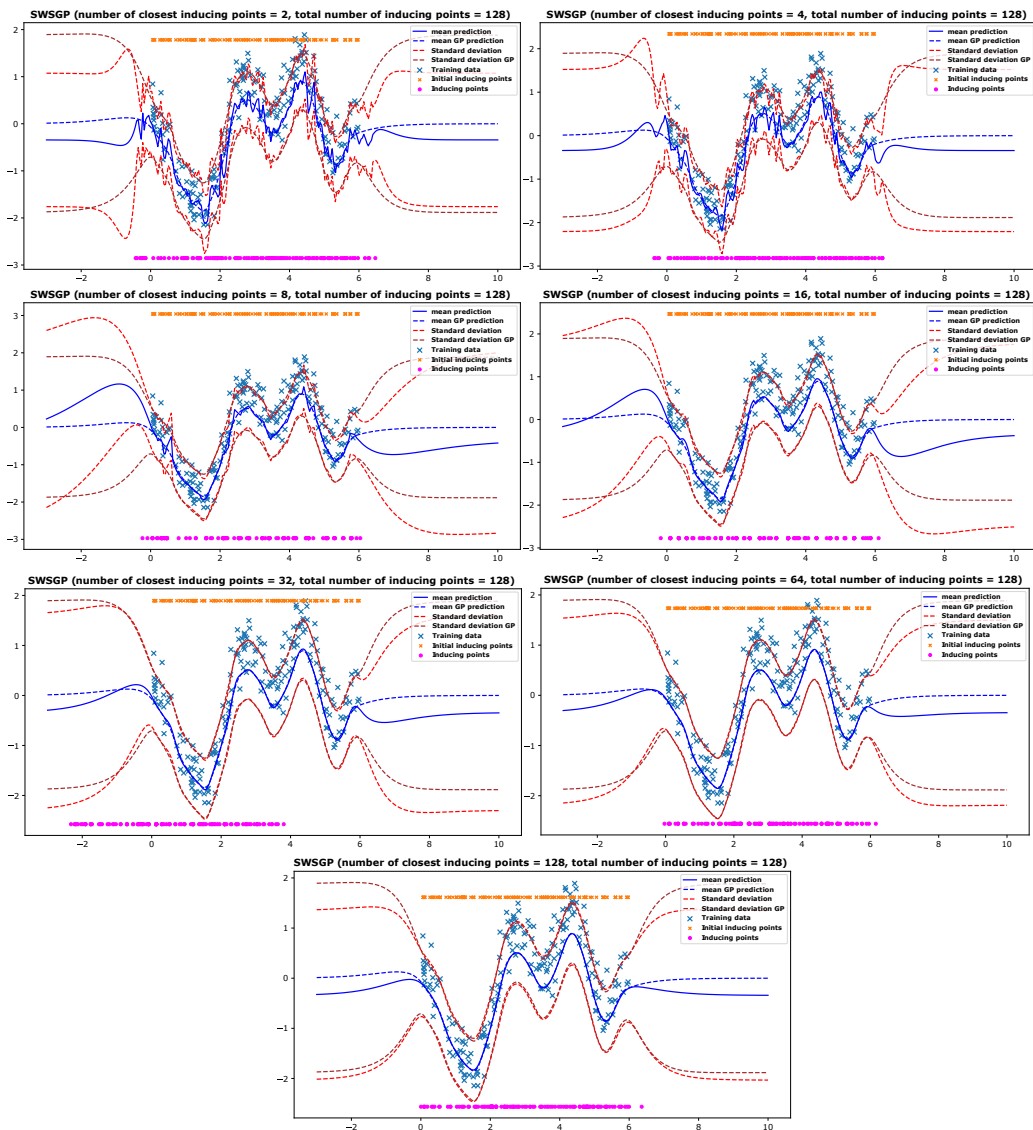

Figure 8: Toy regression example by varying number of the neighbor inducing points $M_c = \{2, 4, 8, 16, 32, 64, 128\}$ and total number of inducing points $M = 128$, with location of initial and final inducing points. The mean and standard deviation of full GP prediction are shown with dashed blue and brown lines, respectively. The blue lines and the dashed red lines are the mean and standard deviation of SWSGP.

## D.2 EXTRA RESULTS FOR THE TOY REGRESSION EXPERIMENT

Here we run the 1D toy regression experiment of Section 5.1 using the closed-form solution approach of Titsias (2009) for finding $q$. More precisely, this method is exactly the same as the SVGP method we compare results with, but where the approximate distribution $q$ is not optimized at all. The reason for this is that it is possible to find a closed-form solution for $q$. However and importantly, the resulting method does not allow for mini-batch training. Since SVGP* does not allow for stochastic optimization, the batch size is set equal to the number of training points ($N = 200$). Figure 9 shows the fit obtained for an increasing number of inducing points $M$. The results are very similar to the ones of SVGP in Figure 7.

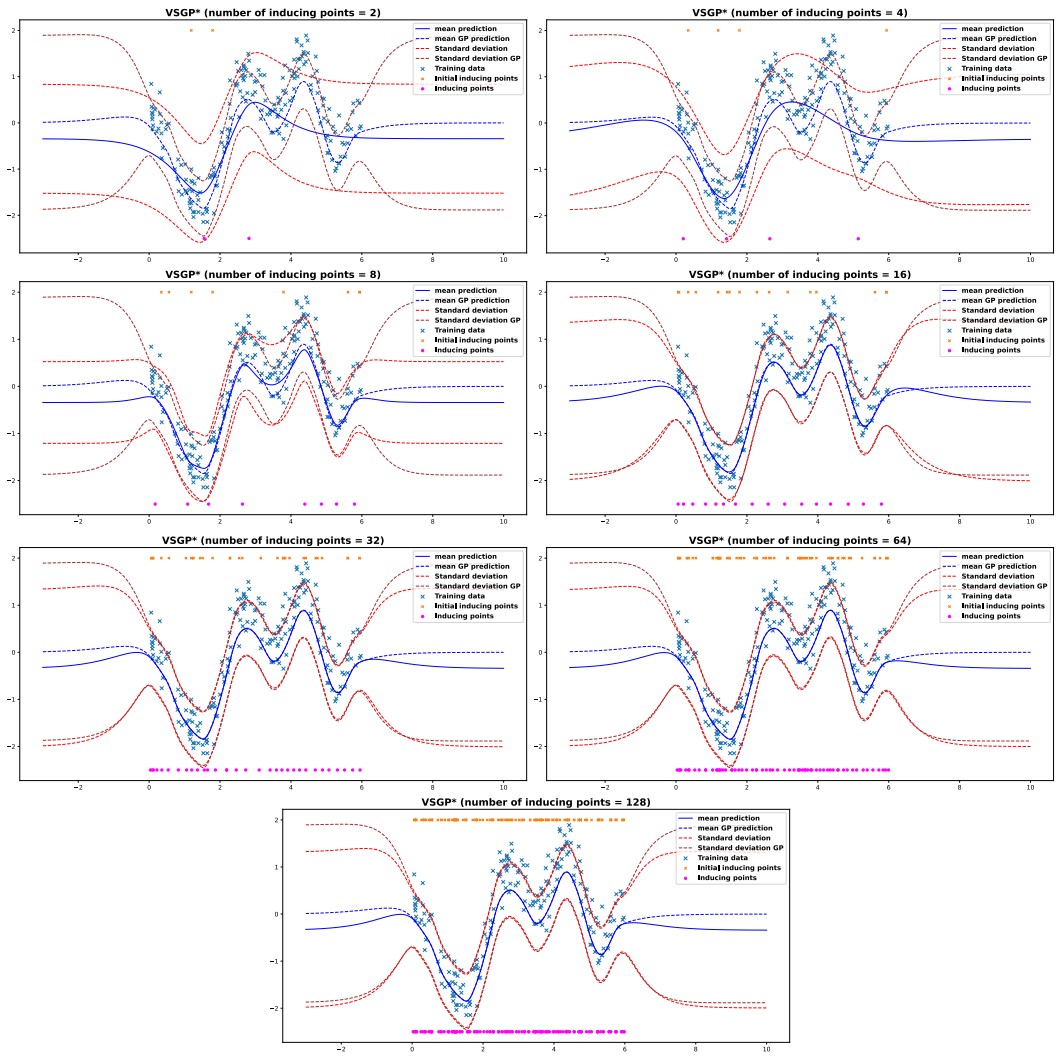

Figure 9: Toy regression example by varying number of inducing points $M = \{2, 4, 8, 16, 32, 64, 128\}$ with location of initial and final inducing points. The mean and standard deviation of full GP prediction are shown with dashed blue and brown lines, respectively. The blue lines and the dashed red lines are the mean and standard deviation of VSGP*.

## D.3 UCI REGRESSION DATASETS

Table 4: Test Root Mean Squared Error (RMSE) values for the UCI regression datasets. The numbers in parentheses are standard errors. Best mean values are highlighted.

|  | $N$ | $d$ | VSGP | SOLVE | SWSGP | IDSGP |
|---|---|---|---|---|---|---|
| Kin40k | 32,000 | 8 | 0.198 (0.002) | 0.157 (0.001) | 0.215 (0.002) | **0.050 (0.002)** |
| Protein | 36,584 | 9 | 4.161 (0.011) | 4.062 (0.011) | 4.133 (0.008) | **3.756 (0.019)** |
| KeggDirected | 42,730 | 19 | 0.032 (0.001) | 0.079 (0.032) | 0.024 (0.000) | **0.022 (0.001)** |
| KEGGU | 51,686 | 26 | 0.024 (0.000) | 0.020 (0.000) | 0.022 (0.000) | **0.014 (0.000)** |
| 3dRoad | 347,899 | 3 | 9.641 (0.063) | 10.020 (0.095) | 11.726 (0.327) | **7.250 (0.069)** |
| Song | 412,276 | 90 | 8.966 (0.022) | **8.925 (0.020)** | 9.013 (0.029) | 9.068 (0.011) |
| Buzz | 466,600 | 77 | 175.076 (15.021) | 173.352 (14.957) | **160.744 (13.467)** | 166.784 (18.040) |
| HouseElectric | 1,639,424 | 6 | 0.035 (0.000) | 0.034 (0.000) | 0.036 (0.001) | **0.032 (0.000)** |
| Avg. Ranks |  |  | 3.075 (0.126) | 2.400 (0.138) | 3.025 (0.170) | **1.500 (0.151)** |
| # Inducing points |  |  | 1024 | 1024 / 1024 | (K=50) / 1024 | 15 |

Table 5: Average prediction time per epoch across the 5 splits for the UCI regression datasets. The numbers in parentheses are standard errors. Best mean values are highlighted.

|  | Kin40k | Protein | KeggDirected | KEGGU | 3dRoad | Song | Buzz | HouseElectric |
|---|---|---|---|---|---|---|---|---|
| VSGP | 0.9(0.00) | 1.1(0.0) | 1.4(0.01) | 1.7(0.01) | 11.6(0.07) | 14.5(0.08) | 18.0(0.08) | 48.3(0.12) |
| SOLVE | 2.4(0.00) | 2.8(0.0) | 3.2(0.00) | 4.0(0.00) | 27.0(0.04) | 31.8(0.19) | 37.0(0.03) | 127.7(0.43) |
| SWSGP | 1.3(0.00) | 1.5(0.0) | 1.8(0.01) | 2.1(0.01) | 14.8(0.03) | 17.6(0.02) | 21.3(0.10) | 66.2(0.88) |
| IDSGP | **0.3(0.00)** | **0.5(0.0)** | **0.7(0.00)** | **1.0(0.02)** | **5.7(0.26)** | **5.6(0.05)** | **8.1(0.08)** | **22.1(0.14)** |

## D.4 UCI CLASSIFICATION DATASETS

Table 6: Test Accuracy values for the UCI classification datasets. The numbers in parentheses are standard errors. Best mean values are highlighted.

|  | $N$ | $d$ | VSGP | SOLVE | SWSGP | IDSGP |
|---|---|---|---|---|---|---|
| MagicGamma | 15,216 | 10 | 0.876 (0.001) | 0.877 (0.002) | 0.867 (0.002) | **0.877 (0.002)** |
| DefaultOrCredit | 24,000 | 30 | **1.000 (0.000)** | **1.000 (0.000)** | **1.000 (0.000)** | 1.000 (0.000) |
| NOMAO | 27,572 | 174 | 0.956 (0.002) | 0.960 (0.001) | **0.961 (0.001)** | 0.955 (0.001) |
| BankMarketing | 36,169 | 51 | 0.906 (0.001) | **0.907 (0.001)** | 0.897 (0.001) | 0.905 (0.001) |
| Miniboone | 104,051 | 50 | 0.941 (0.001) | **0.945 (0.001)** | 0.938 (0.000) | 0.937 (0.001) |
| Skin | 196,046 | 3 | 0.999 (0.000) | 0.999 (0.000) | 0.999 (0.000) | **0.999 (0.000)** |
| Crop | 260,667 | 174 | 0.999 (0.000) | 0.999 (0.000) | **0.999 (0.000)** | 0.999 (0.000) |
| HTSensor | 743,193 | 11 | 0.999 (0.000) | **1.000 (0.000)** | 0.989 (0.003) | 0.999 (0.000) |
| Avg. Ranks |  |  | 2.475 (0.127) | **1.975 (0.152)** | 2.900 (0.187) | 2.650 (0.168) |
| # Inducing points |  |  | 1024 | 1024 / 1024 | (K=50) / 1024 | 3 |

Table 7: Average training time per epoch across the 5 splits for the UCI classification datasets. The numbers in parentheses are standard errors. Best mean values are highlighted.

|  | Magic | DefaultOrCredit | NOMAO | BankMarket | Miniboone | Skin | Crop | HTSensor |
|---|---|---|---|---|---|---|---|---|
| VSGP | 3105(459) | 4759(516) | 4445(549) | 6231(862) | 18447(1279) | 37835(7065) | 49962(9292) | 115463(17511) |
| SOLVE | 5154(1061) | 7554(1039) | 6718(1028) | 8949(1635) | 37022(7902) | 64606(13314) | 88819(18864) | 168709(21194) |
| SWSGP | 1547(145) | 2354(182) | 2728(188) | 3682(351) | 10040(347) | 20283(2796) | 21770(3038) | 67687(5880) |
| IDSGP | **1143(100)** | **1293(90)** | **2026(94)** | **2987(354)** | **7654(134)** | **15700(1918)** | **21378(2561)** | **53895(5652)** |

Table 8: Average prediction time per epoch across the 5 splits for the UCI classification datasets. The numbers in parentheses are standard errors. Best mean values are highlighted.

|  | MagicGamma | DefaultOrCredit | NOMAO | BankMarketing | Miniboone | Skin | Crop | HTSensor |
|---|---|---|---|---|---|---|---|---|
| VSGP | 3.6(0.56) | 4.1(0.59) | 4.4(0.74) | 9.5(4.39) | 17.9(1.84) | 49.7(11.15) | 58.7(12.82) | 139.7(41.67) |
| SOLVE | 4.0(0.85) | 5.4(0.73) | 3.4(0.76) | 4.9(1.24) | 49.2(17.83) | 48.8(16.73) | 86.8(36.26) | 93.5(15.73) |
| SWSGP | 3.0(0.43) | 3.9(0.38) | 4.3(0.51) | 5.4(0.72) | 16.4(1.82) | 36.0(8.00) | **33.9(6.25)** | 88.4(9.16) |
| IDSGP | **2.5(0.24)** | **2.5(0.21)** | **3.5(0.39)** | **4.8(0.54)** | **13.9(0.75)** | **26.1(4.92)** | 37.5(4.96) | **83.4(8.23)** |

## D.5 LARGE SCALE DATASETS

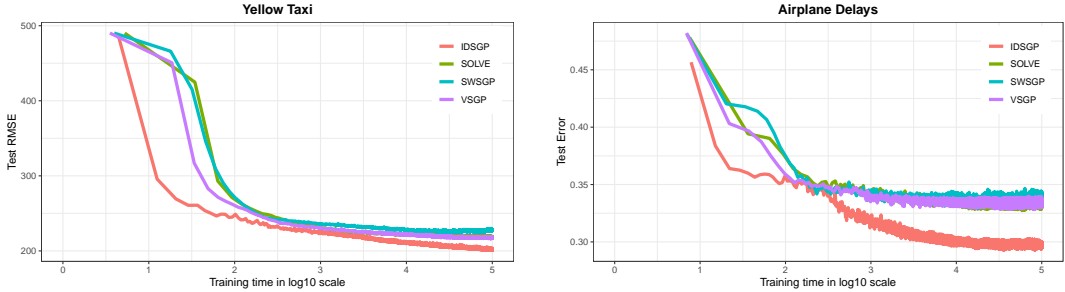

Figure 10: (left) Test RMSE for each method as a function of the training time in seconds, in $\log_{10}$ scale, for the Yellow taxi dataset. (right) Prediction error on the test set for each method as a function of the training time in seconds, in $\log_{10}$ scale, for the Airlines Delays dataset. Best seen in color

## D.6 NEURAL NETWORK TRAINED VIA MAXIMUM LIKELIHOOD

In this subsection we show extra experiment results on the UCI datasets when using a neural network trained via maximum likelihood. The architecture of the neural network is the same as the one of the network used in the proposed method IDSGP. Training is done using ADAM. The learning rate used is 0.001. The mini-batch size is the same for the GP-based methods. In regression, we use the neural network to predict the mean and variance of the Gaussian predictive distribution. In classification, we use a sigmoid activation function. The average test negative log-likelihood obtained in each problem is shown in Table 9 and Table 10. The results are high-lighted in bold-face when the NN performs worse than any other GP based method. The tables show that, in the case of regression problems, most of the times the neural network performs worse than the GP based methods. In the case of classification problems, the performance of the neural network is worse in those problems in which the error is higher according to Table 6. By contrast, in those problems in which the accuracy is almost equal to 100%, there are no differences or it performs slightly better.

Table 9: Avg. neg. test log-likelihood values for the UCI regression datasets for the neural network. The numbers in parentheses are standard errors.

| Kin40k | Protein | KeggDirected | KEGGU | 3dRoad | Song | Buzz | HouseElectric |
|---|---|---|---|---|---|---|---|
| **0.099(0.03)** | **2.794(0.02)** | -2.407(0.14) | -5.124(0.18) | **3.661(0.02)** | **3.363(0.01)** | **27.840(33.44)** | -2.110(0.03) |

Table 10: Avg. neg. test log-likelihood values in the UCI classification datasets for the neural network. The numbers in parentheses are standard errors.

| MagicGamma | DefaultOrCredit | NOMAO | BankMarketing | Miniboone | Skin | Crop | HTSensor |
|---|---|---|---|---|---|---|---|
| **0.315(0.02)** | 0.000(0.00) | **0.119(0.01)** | **0.232(0.00)** | **0.152(0.00)** | 0.002(0.00) | 0.002(0.00) | 0.000(0.00) |

