# OpenReview forum: "Input Dependent Sparse Gaussian Processes "
_ICLR.cc/2022/Conference — ICLR 2022 Submitted_

### Official Review · Reviewer_m21i · 2021-10-22

**Correctness:** 3
**Technical Novelty And Significance:** 2
**Empirical Novelty And Significance:** 2
**Recommendation:** 5
**Confidence:** 4

**Main Review:**


## Significant Comments and Questions
*  Suppose I am interested in making a prediction of the joint distribution at a pair of inputs $x,x'$. How would I do this with you method? If I understand correctly, I would feed both $x$ and $x'$ into a neural net, which would output two distinct sets of inducing inputs, as well as distributions over them. Can you clarify how these can be combined to approximate, for example, the posterior covariance between $x$ and $x'$?
* Relatedly, it seems to me as though the posterior predictive distribution $f$ changes between train and test time. In particular, the posterior distribution depends on $p(\tilde{x})$ which differs at train and test time. Do you foresee this leading to any issues?
* Matthews et al 2015 established that maximizing the lower bound in Titsias, 2009 is equivalent to minimizing a KL-divergence between stochastic processes. This is essential to justifying maximization of the lower bound with respect to $Z$ as a form of variational inference. Given the augmented model, it is no longer clear to me that a similar argument applies. I am curious to hear if the authors have any thoughts as to whether it does or does not. If not, does the method risk over-fitting by optimizing $Z$? Relatedly, can the quality of inference (as measured by KL-divergence to the posterior) be shown to be monotonically increasing in $Z$?
* The authors should consider adding the closed form solution for approximate Gaussian process regression in experiments (Titsias, 2009). While more computationally expensive per iteration that the method of Hensman et al 2013, using the analytic form of the optimal posterior often improves inference quality signficicantly, and convergence of the bound proposed in Hensman et al 2013 can be slow.

## Minor Comments and Questions
* The authors repeatedly refer to the "training cost" when they mean the "cost per iteration of training". The training cost should incorporate how long it takes for the training of the method to converge.
* At several points you state that at least several hundred inducing points are needed for complicated problems. It seems as though several thousands of inducing points might be a more appropriate statement. Generally using several hundred inducing points would not represent a major computational obstacle. The lower bounds on the number of inducing points needed  for the KL-divergence between approximate posterior and prior to not be large in Burt et al, 2020 might also be of interest.
* On page two, $k(x,x') = \mathbb{E}[f(x)f(x')]$ is only correct if the mean is $0$. While you state that you generally assume this, it should be more clear this is needed in this definition.
* The last paragraph in section 2.1 is copied almost word for word from an earlier paragraph.
* In experiments, when results are compared, are model parameters selected via ELBO maximization (approximate maximum marginal likelihood)?

## References
``On Sparse variational methods and the Kullback-Leibler divergence between stochastic processes'' Matthews, Hensman, Turner, Ghahramani. 2015.

``Convergence of Sparse Variational Inference in Gaussian Processes
Regression'', Burt, van der Wilk, Rasmussen.

**Summary Of The Paper:**

The paper proposes a variant of the method in Tran, 2021 for approximate Gaussian process regression. The major changes with respect to the classic Titsias, 2009 paper on variational Gaussian process regression are:
* The inducing inputs are data-dependent, in the sense that $Z=Z(x_i)$.
* The inducing inputs, posterior mean, and posterior covariance are learned via amortized variational inference through learning a neural network.

The authors show some promising empirical evidence on commonly used regression and classification benchmarks.


**Summary Of The Review:**

## Summary of Review
The method is interesting and the empirical results seem promising. I do have some concerns that the posterior distribution defined over $f$ is not consistent. Perhaps this is not essential if only marginal predictions are needed but it does seem to be a significant drawback in cases when covariances or samples from the approximate posterior are needed.  This also makes me slightly concerned that the objective function is not quite as well founded as that of Titsias, 2009 (see the discussion in Matthews et al 2015). I think some of these critiques apply equally to Tran et al, and I am curious to hear the authors thoughts on these limitations.

The writing is generally reasonably clear, though aspects could be improved. While I have rated the paper as marginally below the acceptance threshold, I think it is quite close to the threshold.

---

> ### Author Response · Authors · 2021-11-18
> **Response to reviewer m21i**
>
> We would like to thank the reviewer for their valuable comments. We will keep into account all the comments. Below we try respond to the questions raised by the reviewer.
>
> ## Significant Comments and Questions
>
> IDSPG cannot directly be used to compute a multi-variate predictive distribution over several data points simultaneously. Note, however, that this is also a limitation of the method SWSGP we compare favorably with. Such a method suffers from the same limitation as ours. We  have now mentioned this limitation in the manuscript. In any case, most performance and uncertainty quantification metrics do not consider covariances and can be computed without them. Nevertheless, this might be a limitation in applications where covariances are essential. One way to work around this is to consider for prediction the union of input dependent inducing points for test points. However, this would be inconsistent with the training method proposed for IDSGP. Such approach can be modified to take this into account by considering the union of input dependent inducing points for the training points within a mini-batch. This is a solution similar to one proposed in SWSGP for considering covariances.
>
> The neural network is shared between training and testing and not changed. The output of the neural network will depend on the input training data during training and test data during testing. All data are assumed to come from the same distribution. Therefore we do not foresee nor have observed issues about this in our experiments.
>
> First, we think it is not straightforward to carry out a similar analysis as the one performed in Matthews et al 2015 w.r.t. the method of Titsias, 2009. Matthews et al 2015 is a theoretical paper that provided a solid ground on which Titsias, 2009 method was based on. However, Titsias, 2009 method was already popular and showed its utility before Matthews et al. 2015 work existed. We believe that a similar analysis could be possible w.r.t our method. However, that analysis is complicated and we leave it for future work. We agree with the reviewer that it is important to provide solid theory for the machine learning methods being used. However, it is better to provide theory for those methods that have already shown a good empirical utility. On this topic, also note that competing methods such as SWSGP are lacking a similar theoretical analysis. Note also that our method is a generalization of that of Titsias, 2009. In particular, if the DNN outputs always the same set of inducing points and variational parameters for the approximate distribution q, our method, IDSGP, and that of Titsias, 2009 (i.e. SVGP in our manuscript) coincide. Finally, we do not expect our method to over-fit by optimizing Z nor the variational parameters of q. The reason is that the DNN is not a parameter of the model itself (which is a GP) but of the approximate inference algorithm. One can show that the lower bound $\mathcal{L}$, in Eq. (4), satisfies $\mathcal{L} = \log p(\mathbf{y}) - KL(q(\mathbf{f},\mathbf{u},\tilde{\mathbf{x}})|p(\mathbf{f},\mathbf{u},\tilde{\mathbf{x}}|\mathbf{y}))$. Therefore, maximizing $\mathcal{L}$ should decrease the KL-divergence w.r.t. the posterior. We have included this derivation in the appendix.
>
> We haven't considered the closed form solution of Titsias, 2009 because even though it may give better results in some situations, such an approach is not scalable and cannot be applied to very large problems. Our method's main advantage is scalability. By contrast, Titsias, 2009 method is not scalable because it does not allow for mini-batch training and stochastic optimization techniques. Moreover such a method will be also limited by the impact of requiring a large number of inducing points as SVGP.  We have updated the paper and compared in the appendix the quality of the predictive distribution on the toy regression problem. The results obtained, shown in the appendix, are similar to those of SVGP.
>
> ## Minor Comments and Questions
>
> Thanks for noting these points. We have updated the paper to take them into account.
>
> Yes, in the experiments all model's parameters are selected via ELBO maximization in each method.
>
> Please, see the upated paper and a diff.pdf file in the supplementary material showing changes made.
>
> Again, thank the reviewer for the insight analysis and relevant comments. If there any other aspect that needs further clarification we kindly ask the reviewer to indicate so. If there is none, we kindly ask the reviewer to update the given score if all their concerns have been addressed.

---

> > ### Comment · Reviewer_m21i · 2021-11-22
> > **Thanks for the response**
> >
> > Thank you for your response, as well as including a diff of the submission which made sorting through changes much faster. While it is true that Titsias, 2009 predated Matthews, 2015 and was influential even before it was put on more rigorous footing, the less well-established approach in SWSGP that you build off of seems to have some practical issues that arise from not having as strong a theoretical foundation. For example, the lack of a canonical distribution to use to compute covariances. That being said, the empirical results are promising.
> >
> > For the moment, I am leaving my score as it is, but would like to thank the authors for the detailed response.

---

> > > ### Author Response · Authors · 2021-11-23
> > > **Response to reviewer m21i**
> > >
> > > While we have not developed theory for our method, the proposed approach has been shown empirically to outperform competing methods in terms of performance and computational speed, for training and testing (see Tables 5 and 8). Our method provides more accurate predictive distributions when the number of inducing points is fairly small. Thus, we believe that our paper is an important step in the right direction.  We agree that theoretical results are important but they are, in general, difficult to obtain. On the other hand, practitioners are currently using related methods, e.g., SWSGP (also lacking theory) on a daily basis. If we can provide a significantly better and more general method, then we think this is very worthwhile. We hope that the development of useful methods will justify the development of better theory over time; this would be preferable to theory being developed for the less effective methods that exist today.
> > >
> > > Although our method lacks a direct approach to compute covariances, most performance and uncertainty quantification metrics do not consider covariances and can be computed without them. If covariances are strictly needed, the method proposed in the SWSGP paper for obtaining them could be used, as indicated in our response [1]. This only requires slightly changing the training approach. In section 4.5 of [1] it is shown that such a method for computing covariances leads to accurate covariance estimation.
> > >
> > > Again, we thank the reviewer for the thorough and insightful analysis of our paper. We kindly hope that our response can make the reviewer reconsider the decision of keeping the score as it is.
> > >
> > > [1] G.-L. Tran, D. Milios, P. Michiardi, and M. Filippone. Sparse within sparse Gaussian processes using neighbor information, International Conference on Machine Learning, 10369-10378, 2021.

---

### Official Review · Reviewer_QKXW · 2021-10-24

**Correctness:** 3
**Technical Novelty And Significance:** 2
**Empirical Novelty And Significance:** 2
**Recommendation:** 6
**Confidence:** 3

**Main Review:**

The paper has made a trade-off between training speed (pros) and throwing away the readily available gradient from the ELBO (cons).

Strengths:
* Optimizing the ELBO for Sparse GP using deep neural network is interesting for improving the running time (computational cost)

Weaknesses:
* Training a deep neural network may require a large amount of data as opposed to the standard sparse GP.

* It could be a drawback from the design in high-dimensional input setting. This can be inefficient to learn a mapping from one input (in high-dimensional) to multiple output (again in high-dimensional) of the inducing points. Learning such mapping may require a significantly large amount of training data.

* It is unclear about how to make use of the ELBO gradient in learning the neural network. On the one hand, existing variational inference techniques for sparse GP make use of such gradient when taking partial derivative with respect to the variational parameters. On the other hand, with deep neural network, one can utilize the automatic differentiation to perform back-propagation to learn the network (as a black-box). Note that optimizing the ELBO as a black-box (with deep neural network) using automatic differentiation may be inefficient if we have the gradient signal readily available from the ELBO.

Novelty:
* The paper follows Tran et al 2020 to derive the lower bound on the log-marginal likelihood.

* Note that existing works has considered using deep neural network to optimize the ELBO, (but not in the setting of sparse GP), such as [1]

Writing and presentation:
* The paper is well written and nicely presented.
* The related work is great.

Suggestions:
* One of the key advantages of GP against other supervised learning approaches is the uncertainty quantification. Therefore, additionally to presenting the classification and regression performance on UCI datasets, it is recommended to study/analyse/compare the uncertainty estimation. Without such uncertainty estimation, people can simply perform standard deep learning method to learn these classification and regression.


Minor points:
* in the last paragraph of the introduction, the paper claims to have different sets of inducing points associated to each input location. However, given the network, it seems that each input will only produce a single set of inducing point, not a set of including points?

* what is the dimension of Cholesky factor of the covariance matrix S=LL^T and the number of inducing points per data point Z(x_i)

* In Fig 1, there is a typo at the first column, last row which should be h_P^(1), instead of h_P^(L)


[1] Blundell, C., Cornebise, J., Kavukcuoglu, K., & Wierstra, D. (2015, June). Weight uncertainty in neural network. In International Conference on Machine Learning (pp. 1613-1622). PMLR.



**Summary Of The Paper:**

This paper proposes to combine sparse GPs with a neural network architecture to compute the
inducing points locations associated to each input point. In particular, the paper employs a neural network to carry out amortized VI to compute the parameters of the approximate variational distribution
q modeling the posterior distribution associated to the values of the outputted inducing points. The inducing points are given by a mapping from the inputs provided by a neural network. The paper demonstrates improvement on training and prediction time while the proposed method is able to perform similar or better than the standard sparse GP approaches.


**Summary Of The Review:**

The paper proposes to improve the training time and flexibility of sparse GP approximation using deep neural network. While it makes sense to optimize the ELBO using deep neural network, the proposed framework may have certain limitation, such as learning the mapping from one (high-dim) input to multiple (high-dim) output can be inefficient, especially when we have limited data. Another suggestion is to demonstrate the uncertainty estimation of the proposed approach.

The paper is well written and presented. The related work section is well covered.

The paper is currently in the border-line.

---

> ### Author Response · Authors · 2021-11-18
> **Response to Reviewer QKXW**
>
> We would like to thank the reviewer for their valuable comments. We will keep into account all the comments. Below we try respond to the questions raised by the reviewer.
>
> Weaknesses:
>
> Regarding the points considered as weaknesses in the review, it is true that training a DNN may require more data if you compare it to the standard sparse GP. However, in our case the DNN is used as part of the approximate inference mechanism and not as part of the model, so we do not need a large amount of data to train it. As shown in our experiments, our method, IDSGP, provides better results than less flexible methods that do not rely on a DNN such as VSGP and SWSGP. Furthermore, it leads to faster training times.
>
> Regarding the training of the DNN we simply used tensor-flow to implement our model. See the source code provided in the supplementary material. Tensor-flow allows to write a stochastic estimate of the objective where the randomness comes from mini-batch training. This objective is automatically differentiated in Tensor-flow to obtain unbiased noisy gradients. These gradients are then used to optimize the objective via the ADAM optimizer. In our experiments, the proposed method, IDSGP, has shown better computational efficiency than the other methods.
>
> Novelty:
>
> Variational inference and the maximization of the ELBO is not restricted to GP models. Bayesian DNNs can also be trained using variational inference and the ELBO as the loss function. This is done in [1], as indicated by the reviewer. However, our method is different from that work in the sense that we use use a GP as the underlying model and not a deep neural network. In our case, the DNN is part of the approximate inference algorithm. More precisely, it is used to find a non-linear transformation between the inputs and the inducing points and to amortize the computation of the variational parameters.
>
> Suggestions:
>
> We already perform experiments evaluating the quality of the uncertainty estimation. In particular, in the synthetic regression problems we plot the predictive distribution (mean and standard deviation), which is shown to be more accurate for our method. Importantly, in the other experiments we report the test log likelihood of each method. This quantity takes into account the quality of the whole predictive distribution, which considers also prediction uncertainty. Since the negative test log likelihood of our method is often better, this indicates a better predictive distribution in general and a more sensible uncertainty estimation.
>
> Minor points:
>
> - We would like to apology for not expressing this correctly in the manuscript. We meant that for each input location we will have a set of inducing points specified by the output of the DNN. This means that the set will be different depending on the input of the DNN, which will be different for each input location. We have rewritten this last paragraph.
>
> - The Cholesky factor of the covariance matrix S is a lower triangular matrix of size M x M, where M is the number of inducing points per data point. Each set of inducing points Z(x_i) has size M x D, where D is the dimensionality of the problem.
>
> - Thank you for noting the typo in Figure 1, we have corrected it in the manuscript.
>
> Please, see the upated paper and a diff.pdf file in the supplementary material showing changes made.
>
> Again, thank the reviewer for the insight analysis and relevant comments. If there any other aspect that needs further clarification we kindly ask the reviewer to indicate so. If there is none, we kindly ask the reviewer to update the given score if all their concerns have been addressed.
>
> References:
>
> [1] Blundell, C., Cornebise, J., Kavukcuoglu, K., & Wierstra, D. (2015, June). Weight uncertainty in neural network. In International Conference on Machine Learning (pp. 1613-1622). PMLR.

---

> > ### Comment · Reviewer_QKXW · 2021-11-26
> > **response read**
> >
> > I would like to thank the authors for the detailed response. I will keep my original score.

---

### Official Review · Reviewer_rpPD · 2021-11-02

**Correctness:** 3
**Technical Novelty And Significance:** 2
**Empirical Novelty And Significance:** 2
**Recommendation:** 3
**Confidence:** 3

**Main Review:**

The main contribution of the paper is using a deep neural network (DNN) to find inducing points' locations and variational distribution parameters. In general form, it can reduce the computational cost of the original methods (SWSGP and SVGP) by training the variational parameters as the output layer of a DNN. But such a reduction in computational cost happens when it uses a small number of inducing points for each data point and small network architecture. However, there are concerns about the paper:

1- Novelty and Contribution: The paper has been inspired by the SWSGP method Tran et al. (2020). SWSGP samples a mini-batch from the input data set $\mathcal{D}$ and for each data point in the mini-batch, it finds H nearest inducing points. Instead of finding the nearest inducing points, IDSGP uses a deep Neural network and amortized variational inference Kingma and Welling (2014) and Shu et al. (2018) (available method) to find a small number of inducing points for each data point in mini-batch. The other parts of the algorithm in IDSGP are the same as the proposed algorithm in Tran et al. (2020). Indeed, using DNN to estimate variational parameters is not a novel approach. For instance, this network architecture is used in the encoder part of the conventional variational auto-encoder (VAE) (and also variational inference with the normalizing flow) where the parameters of the variational distribution (and flow parameters in normalizing flow) are estimated via DNN.

 2-  Limitations of the Method: The limitations of the method are not properly discussed. The objective function has severely pathological properties which have not been described in the paper. For instance, consider the $n=M=1$ case, where n is the batch size, and a stationary kernel with unit variance. The optimized bound is exactly equal to a sum of GPs with one point each i.e. $\sum_i N(y_i, | 0, 1 + \sigma^2)$, and this does not depend on the prior at all. In fact,  as all priors are equivalent to the white noise prior. The SWSGP method does indeed have a similar problem, considering the $n=M=H=1$, but in the paper, it is explicitly described how the effective prior is modified.

3- Fall back to the Prior:  In regions of the space without training data the extrapolation is determined by the DNN and there is no guarantee the predictions fall back to the prior. This is actually apparent in Figure 2 where VSGP is guaranteed to give the prior when there is no data or inducing points, whereas IDSGP visibly extrapolates in a way that doesn't seem sensible.

4- Performance: It is not clear why the IDSGP method can provide better prediction quality than SWSGP or SVGP while it only uses a small number of inducing points. Why is the DNN is better than the nearest point sampling in SWSGP, given that they maximize the same lower bound? SWSGP uses some nearest inducing points for each data point while IDSGP uses a small set of inducing points that have been estimated for each data point. Why does the second one have better performance? Since H-inducing points in SWSGP are selected based on input points, SWSGP is also an input-dependent SGP method. There is no reason to accept the inducing point set (in IDSGP) can lead to better predictive posterior and its improvement in prediction quality is ambiguous. For instance, in Appendix D.1, when the number of nearest inducing points (H) increases, SWSGP presents better results and its uncertainty is close to the standard GP.

 5- Computational cost: The paper claims that since it uses a fewer number of inducing points, computational cost can be improved. However, it is not always true and the paper has not presented a related analysis about that. For instance, let consider the toy example in Section 5.1 and Figure 2. In this small data set, SWSGP is a little faster than IDSGP based on the computational cost discussions in both papers and experiment settings (i.e. the number of the layers and hidden units in DNN, number of inducing points in IDSGP, and the value of H and M in SWSGP). Besides, for a non-diagonal variational covariance matrix S, the output layer of DNN has $2M + \frac{M^2+M}{2}$ variables (M for inducing points, M for mean and $\frac{M^2+M}{2}$ for Cholesky matrix). When it needs more inducing points for each input point, the computational cost of DNN increases because of the dimension of the output layer and also network parameters (number of layers and hidden units).



**Summary Of The Paper:**

Paper considers the sparse approximation Gaussian processes (GPs) which tries to scale GPs for large data sets. The authors in this paper propose a novel method (IDSGP) to improve the computational cost of sparse GPs. With the proposed method, the number of the inducing points is reduced drastically and only a small number of inducing points are estimated for each input point. The inducing points locations, as well as the parameters of the variational posterior distribution, are the outputs of a Deep neural network (DNN).

First, they sample a mini-batch of the training data. For each data point in the mini-batch, their solution produces a small set of inducing points (and associated variational approximation) that are specific to each input data point.

It is an extension for Sparse within a Sparse GP (SWSGP) method Tran et al. (2020). For M inducing points, the SWSGP operates on a subset of H inducing points for each input point, with $H \ll M$, while maintaining a sparse approximation with M inducing variables. Here, instead of H nearest inducing points, the model estimates M inducing points and variational distribution parameters in a DNN. The claim is the proposed IDSGP method can provide competitive predictions while it improves the time complexity of the state-of-the-art sparse GPs baselines.

**Summary Of The Review:**

In general, the paper is well-written. However, the proposed model has not been explained appropriately in Section 3.

Section 3.2 is the main contribution of the paper which proposes the computational graph that estimates inducing points and variational parameters. Here it is necessary that the authors address the major issues of their method: why maximizing the lower bond (Equation (4)) in the proposed DNN leads to better results than the available baselines (e.g. SWSGP and SVGP which maximize the same objective function)?  Why does a small number of inducing points provide a better posterior than other baselines which find more inducing points (for instance SWSGP provides more inducing points but uses only H nearest inducing points for each input point)? Indeed, it is not clear how the batch permutation-invariance is achieved.  Something like a sum over the batch dimension is clearly not a good idea because the encoder is sensitive to the ordering of the batch. However, the paper does not mention how this issue is considered.

Besides, here are some omissions of related work that are not much discussed but could be mentioned, e.g.
https://arxiv.org/pdf/1303.0383.pdf,
https://mlg.eng.cam.ac.uk/zoubin/papers/aistats07localGP.pdf,
https://www.dbs.ifi.lmu.de/~tresp/papers/bcm6.pdf,
http://proceedings.mlr.press/v84/salimbeni18a.html.

---

> ### Author Response · Authors · 2021-11-18
> **Response to Reviewer rpPD**
>
> We would like to thank the reviewer for their valuable comments. We will keep into account all the comments. Below we try respond to the questions raised by the reviewer.
>
> [Novelty and Contribution] Our main contribution is using a DNN to amortize variational parameters and inducing points. We show that such a formulation can reduce the training cost and also improve performance. SWSGP uses a nearest neighbor approach for finding inducing points and does not benefit from a deep neural network. We agree with the reviewer that amortization is not new, and we mention that in the related work section. However, the use of amortized variational inference in the context of sparse GPs based on inducing points is a novel contribution. We believe such a contribution is highly relevant since it leads to better sparse GP results than those of already existing methods at a lower cost, as shown by our experiments. The reason for this is the extra flexibility of amortizing the inducing points, which results in input dependent inducing points. Summing up, we believe that such a result is interesting and novel enough to receive the attention of the community.
>
> [Limitations of the Method n=M=1] We do not see why setting n = M = 1 results in a pathological case. We kindly ask the reviewer to provide more details about this. The batch size does not change the objective that is being optimized. It will only have an impact on the noise of the gradients. If M = 1, the prior will still have an impact on the model via the KL term in (5). That term will force the parameters of q to be close to the prior, i.e., a standard Gaussian.
>
> [Fall back to the Prior] IDSGP is not guaranteed to predict something similar to the prior in regions with no data. However, Fig. 2 shows that the predictive uncertainty of IDSGP is higher in regions with no data, which is the expected behavior. Furthermore, its predictive mean becomes closer to the prior in those regions. Moreover, Fig. 5, in the appendix, shows that IDSGP becomes more similar to the full GP as M increases. Table 1 also shows that IDSGP obtains better test log-likelihoods than the other methods. Thus, its predictive distribution is more accurate. Summing up, we believe that the lack of guarantees of predicting something similar to the prior in regions with no data is compensated by the good empirical results of IDSGP.
>
> [why the IDSGP method is better than SWSGP] SWSGP is also an input dependent method. The better results of IDSGP can be explained because the NN is expected to be more flexible. In particular, SWSGP can only output inducing points contained within the set of M total inducing points. By contrast, if the DNN is flexible enough, it can output any arbitrary location for the inducing points. We agree that SWSGP becomes more accurate as H is increased. But the same happens in IDSGP when the number of inducing points is increased. See Fig. 5. Importantly, when the number of inducing points considered is fairly small IDSGP gives better results than SWSGP, as shown in our experiments.
>
> [Computational cost] A large number of inducing points in our method will incur in extra cost due to the need of using the DNN. However, DNNs usually involve matrix operations that are highly optimized by software for linear algebra. Moreover, those operations (matrix multiplications) can be highly sped-up in a GPU. By contrast, SWSGP requires finding the nearest neighbors (which is expensive), an operation that cannot use the benefits of highly optimized software for linear algebra. Also, the cost of SWSGP depends linearly on the total number of inducing points M, which can be rather large, and much larger than H. E.g., in Table 1, M=1024 for SWSGP, while IDSGP only uses 3 inducing points.
>
> [Section 3.2] IDSGP performs better due to the extra flexibility of the DNN that allows to have input dependent inducing points. Moreover, these input dependent inducing points are more flexible than those of SWSGP for the reasons explained before. Batch permutation invariance is achieved because the output of the neural network will always be the same for the same input. That means that if one randomly permutes the instances within a batch, the output of the DNN will be similarly permuted.
>
> [omissions of related work] We thank the reviewer for these references. We have included them now in our paper. However, note that even though these works are related to sparse GP approximations, they significantly differ from our proposed approach.
>
> Please, see the upated paper and a diff.pdf file in the supplementary material showing changes made.
>
> Again, thank the reviewer for the insight analysis and relevant comments. If there any other aspect that needs further clarification we kindly ask the reviewer to indicate so. If there is none, we kindly ask the reviewer to update the given score if all their concerns have been addressed.

---

> > ### Comment · Reviewer_rpPD · 2021-11-26
> > **Response**
> >
> > I would like to thank the authors for their response. For the moment, I am leaving my score as it is.

---

### Official Review · Reviewer_spvN · 2021-11-02

**Correctness:** 3
**Technical Novelty And Significance:** 3
**Empirical Novelty And Significance:** 2
**Recommendation:** 3
**Confidence:** 4

**Main Review:**

1) Strengths
 - a) The manuscript is rather well written and simple to understand.
 - b) The building blocks are established and well-understood.
 - c) The evaluation is done on relevant public datasets.
 - d) Code is available.

2)Concerns
 - a) It remains open which properties of the GP framework used in the motivation for the work remain intact after all.
 - b) The empirical evaluation is lacking at least three simple baselines.
     i) A neural network predicting the class probability rather than the variational parameters with sigmoid activation in the last layer. Along the same line, a regression network predicting mean and standard deviation (or log standard deviation). Both networks should be trained with maximum likelihood.
     ii) A sparse GP using FITC approximation.
     iii) A neural network predicting the inducing points rather than the variational parameters which is then used in combination with FITC or similar.

3) Remarks
 - a) The Abstract states that "for some learning tasks a large number of inducing points may be required" and that the proposed method is going to address this point. However, the conclusion on the "drastically reduced" number of inducing points is a little unfair as one could argue that all the complexity is shifted into the network for inducing point prediction. One should rather worry about number of parameters instead.
 - b) I have the impression that a direct neural network prediction of the class probability or the mean/std (possible mean/log(std)) would do a similar job as IDSGP and that the training would be much faster.

**Summary Of The Paper:**

The manuscript proposes a variant of a sparse variational approximation in order to render Gaussian process inference scalable. The central idea of the work is to use variationa inference and have a neural network predict the parameters of the variational distribution directly from the input data rather than optimising them using ELBO. Experiments on several substantial regression and classification datasets conclude the paper. The available code was not checked for this review.

**Summary Of The Review:**

The proposed algorithm is certainly an option that has not been explored before but the practical value is at least questionnable due to the missing baseline experiments.

---

> ### Author Response · Authors · 2021-11-18
> **Response to Reviewer spvN**
>
> We would like to thank the reviewer for their valuable comments. We will keep into account all the comments. Below we try respond to the questions raised by the reviewer.
>
> [properties of the GP framework] As mentioned in the paper, our formulation does not change GP prior for the process at the training data. See Eq. (3). Our experiments also show in Fig. 5 that, as M increases, IDSGP becomes closer to the full GP, as expected. The good results obtained by IDSGP in Table 1 also indicate that it produces a more accurate predictive distribution than the other methods. Fig. 1 also shows that in regions with no data the prediction uncertainty of IDSGP increases and the mean is closer to the prior, as in a standard GP.
>
> [regression and classification neural networks] First, we kindly point out that the focus of our paper is improving GPs sparse approximations. These methods are scalable and naturally produce a predictive distribution accounting for epistemic and aleatoric uncertainty. Standard neural networks do not provide such a distribution. They will only output point estimates for the target, in the case of regression. In the case of classification, they will output class probabilities. However, these probabilities only depend on the distance to the decision boundary of the linear hyper-plane of the last layer. Thus, they do not consider epistemic uncertainty. This means that the neural network can be very confident in regions with no data. This is described in [1] (see Sec. 1.5 there). This is expected to lead to poor predictive distributions and low test log-likelihood. Third, in the case of regression, it is not clear how to directly predict the standard deviation. One can predict the variance of the additive noise, but again that is not expected to generate large variances in regions with no data, leading to poor test log-likelihoods. We evaluate these approaches in the appendix confirming that they give worse results, in general, than GP-based methods.
>
> [FITC approximation] We already compare with a sparse GP based on VI, SVGP [3,4]. A detailed comparison between FITC and VI is done in [2]. There, is is shown that the VI objective is more accurate, often works better and it is hence recommended over FITC. Critically, FITC, as described in [5], does not lead to an objective with a sum across data points. Thus, such an approach cannot be easily combined with mini-batch training and cannot scale to large problems, unlike SVGP.
>
> [predicting only induicing points] This is infeasible. In our formulation, for each input, we may have different inducing points, due to the neural network. Potentially infinite different since the output of the neural network will change smoothly during training. Critically, each inducing point z_i has associated a process value u_i=f(z_i). Therefore, we should compute a posterior distribution for each of these latent process values, which are potentially infinite. The only feasible way of doing that is to amortize also the variational parameters of q using a neural network, making them also input dependent, as we do in IDSGP. Finally, it is not clear how to combine a neural network that computes input dependent inducing points with the FITC approximation. That could constitute a different research work. Moreover, even if such a method was feasible, the FITC approximation is expected to suffer from the limitations described above.
>
> [number of parameters] It is true that the number of parameters of our method is increased due the use of the neural network. Our experiments, however, show that such an approach results in not only better performance, but also faster training. See, e.g., Table 1 and 3. In any case, we have included in the introduction a sentence saying that the number of parameters of the method may increase.
>
> [direct neural network] We have evaluated such a method in the appendix. It lead worse results, probably for the reasons given above.
>
> Please, see the upated paper and a diff.pdf file in the supplementary material showing changes made.
>
> Again, thank the reviewer. If there any other aspect that needs further clarification we kindly ask the reviewer to indicate so. If there is none, we kindly ask the reviewer to update the given score if all their concerns have been addressed.
>
> [1] Y. Gal.  Uncertainty in Deep Learning. PhD Thesis. University of Cambridge, 2016.
> [2] Bauer, Matthias, Mark van der Wilk, and Carl Edward Rasmussen. "Understanding probabilistic sparse Gaussian process approximations." In NIPS, 2016.
> [3] M.K. Titsias. Variational learning of inducing variables in sparse Gaussian processes. In AISTATS, pp. 567–574, 2009.
> [4] J. Hensman, N. Fusi, and N. D. Lawrence. Gaussian processes for big data. In UAI, pp. 282290, 2013.
> [5] E. Snelson and Z. Ghahramani. Sparse Gaussian processes using pseudo-inputs. In NIPS, pp. 1257–1264, 2006.

---

> > ### Comment · Reviewer_spvN · 2021-11-24
> > **Empirical results and revision**
> >
> > Many thanks for adding the experiments with the simple baseline **Direct NN** trained with maximum likelihood to the appendix.
> >
> > Looking at Tables 1+9, the simple baseline **Direct NN** is on par with IDSGP on Protein/KeggDirect and better than IDSGP on KEGGU, Song, HouseElectric.
> >
> > Looking at Table 2+10, the simple baseline **Direct NN** is better most of the time and never worse.
> >
> > Hence, it seems, the simple baseline is extremely competitive with the proposes method IDSGP.
> >
> > As my interpretation is distinctively different from your conclusion *"We have evaluated such a method in the appendix. It lead worse results, probably for the reasons given above"*.
> >
> > I'm keeping my score as the added complexity of the paper does not improve results over a simple and extremely fast baseline approach (not mentioned in the paper).

---

> > > ### Author Response · Authors · 2021-11-27
> > > **Response to the comments about the empirical results and revision of our paper**
> > >
> > > We would like to thank the reviewer for providing additional insightful comments about our rebuttal.
> > >
> > > While the direct NN may provide similar test log-likelihoods to our proposed method, the quality of the predictions in terms of the RMSE is worse. We show the results obtained here:
> > >
> > > Regression:
> > >
> > > DataSet		Kin40k	Protein	KeggDirected	KEGGU	3dRoad	Song	Buzz	HouseElectric
> > >
> > > RMSE		0.267	4.452	0.151		0.051	12.275	8.798	183.161	0.047
> > >
> > > RMSE_std	0.01          0.03	0.06		        0.00	         0.14	0.05	         28.56	0.00
> > >
> > > In classification problems, however, the accuracy is similar. The results obtained are:
> > >
> > > Classification:
> > >
> > > Avg. Accuracy values and their standard deviations for the UCI regression datasets with the neural network
> > >
> > > DataSet	 MagicGamma	DefaultOrCredit	NOMAO	BankMarketing	Miniboone	Skin	         Crop	HTSensor
> > >
> > > Accuracy		0.875	1.000		       0.961		0.900	           0.940		1.000	 1.000	1.000
> > >
> > > Accuracy_std	0.01		0.00		                0.00		         0.00	            0.00		    0.00	  0.00	0.00
> > >
> > > In any case, a direct NN approach is expected to be limited by the behavior described in our initial response. Namely, a direct NN does not consider epistemic uncertainty. This means that the neural network can be very confident in regions with no data, which is contrary to what one should expect. GP-based approaches do not have this problem since, in regions far from the observed data, the predictive distribution is expected to be similar to the prior. Falling back to the prior in those regions happens in GPs because the covariances are expected to be small since the observed data is far away.
> > >
> > > For this reason, a direct NN approach is expected to underestimate the predictive variance in in-between data. That is, the predictive uncertainty of a direct NN approach in a region between two clusters of observed points is expected to be fairly small, leading to too confident predictions. See [1,2] for further details about in-between data uncertainty.
> > >
> > > Standard train/test splits are not adequate to estimate the quality of the predictive distribution in regions with no data, as indicated in [1,2]. Gap splits are preferred. In gap splits one sorts the data across each dimension (there is one split per dimension) to then use for training the first 1 / 3 of the instances and the last 1 / 3 of the instances. The middle 1 / 3 of the instances are left aside for testing.
> > >
> > > We have evaluated the direct NN approach and our method, IDSGP, on the Energy dataset extracted from the UCI repository using gap splits. It is well known that this dataset requires sensible in-between data uncertainty estimation to get good prediction results when using gap splits. See [1,2] for further details. The results obtained for the RMSE and the negative test log-likelihood of each method are:
> > >
> > > Method		IDSGP		direct NN
> > >
> > > RMSE		5.11745		4.794854
> > >
> > > RMSE_std	1.376966	        1.217728
> > >
> > > Neg. LL		3.567727	        167.497
> > >
> > > Neg. LL_std	0.6879157	105.8062
> > >
> > > We observe that the direct NN approach and IDSGP have similar RMSE test values when using gap splits. However, the negative test log-likelihood of the direct NN approach is much worse. Because the RMSE is similar, the explanation is that the direct NN approach is providing a less sensible estimation of the prediction uncertainty. An initial exploration shows the predictive variance of the direct NN is smaller for in-between data instances. This confirms that the direct NN approach underestimates the predictive variance in regions with no observed data.
> > >
> > > We will update the paper to include these extra experiments in the supplementary material.
> > >
> > > Again, thank the reviewer. If there is any other aspect that needs further clarification we kindly ask the reviewer to indicate so. If there is none, we kindly ask the reviewer to update the given score if all their concerns have been addressed.
> > >
> > > [1] - Morales-Alvarez, P., Hernández-Lobato, D., Molina, R., & Hernández-Lobato, J. M. (2020, September). Activation-level uncertainty in deep neural networks. In International Conference on Learning Representations.
> > >
> > > [2] - A.Y.K. Foong, Y. Li, J.M. Hernández-Lobato, and R.E. Turner. ’in-between’ uncertainty in Bayesian neural networks. ICML 2019 Workshop on Uncertainty and Robustness in Deep Learning, 2019.

---

### Official Review · Reviewer_oyvg · 2021-11-03

**Correctness:** 3
**Technical Novelty And Significance:** 4
**Empirical Novelty And Significance:** 4
**Recommendation:** 8
**Confidence:** 3

**Main Review:**

I liked the paper a lot once I understood what was going on.  But even with some familiarity with variational GPs, it took me a while to process this.  Most of my comments are individually minor, but they share a common theme that they are not easy to process without either reading out of order or knowing something about variational GPs already.  I think the idea here is quite interesting, and hope thinking about these points will make the paper more accessible!

Detailed comments:

- Top of p2: "This reduces the training cost to O(M^3)" -- perhaps say "O(M^3) per iteration"?

- The kernel is introduced as k(x,x') = E[f(x) f(x')], but perhaps it would be better to write E[(f(x)-m(x)) (f(x')-m(x'))] since the mean function is only "often set to zero" at this point?

- On page 2, we have "a popular covariance function k(.,.) is the squared exponential", but a Matern 3/2 kernel is used later in the experiments.  Perhaps it would be better to replace "squared exponential" with "Matern family" or the like?

- "In Titsias (2009), they do optimize q(u) in closed-form" -> "In Titsias (2009), they optimize q(u) in closed form"

- "The ELBO can be expressed as a sum..." -- there is no prior explicit reference to the evidence lower bound, though of course the bound itself has already been introduced.  It's probably worth a word or two here.

- "To achieve this, we consider a meta-point xtilde" -- I still don't completely understand the role of the meta-points (or what is meta about them)!  Some intuition would be useful.  Also, it is unfortunate that the meta-points are quickly given the same notation as the data points.  It seems like they end up coinciding in the algorithm as implemented, but are conceptually distinct from the data points?

- In Algorithm 1, I think standard fonts (vs the math spacing) would be better for ELBO, KL_div, and Log_lk.

- In related work: "it is introduced a new interpretation" -> "the authors introduce a new interpretation"; "that allow to consider" -> "that allow one to consider"; "it is also described a mechanism" -> "a mechanism is also described".

- As a general comment: the approximation for the pointwise posterior variance is clear, and this is the most important part of the uncertainty in many cases.  But it's unclear to me how one would compute the approximate posterior covariance between two points, which is also sometimes useful.

**Summary Of The Paper:**

The paper describes a method that combines variational Gaussian processes with a neural network that produces inducing points and associated variational parameters depending on the evaluation location.  The approach reduces training time compared to other sparse GP approaches, and has excellent performance in regression experiments (and good performance in classification experiments).  The method also does a good job at respecting the (exact) GP prior structure on the latent functions.

**Summary Of The Review:**

Though some points of the presentation could be improved, I think this is an interesting idea illustrated by good experience.  I would like to see it appear in the conference.

---

> ### Author Response · Authors · 2021-11-18
> **Response to Reviewer oyvg**
>
> We would like to thank the reviewer for their valuable comments. We will keep into all the comments for the camera-ready version of the paper in case of acceptance. Below we will try respond to the main comments as well as the questions raised by the reviewer.
>
> In general, we appreciate the positive points indicated by the reviewer.
>
> [detailed comments] We have addressed the comments and revised them in the manuscript.
>
> [meta-point xtilde] In our model specification, meta points need not correspond to the observed data points. A meta-point is a sample drawn from an arbitrary implicit distribution p(\tild{x}). However, in practice, in our method and, after the amortization for mini-batch optimization, we let that arbitrary distribution be the one corresponding to the observed data. Therefore, we use training data points in practice. This results in an input dependent sparse GP with good generalization properties, as shown in our experiments, and extra flexibility in the sense that less inducing points are needed to get good predictions.
>
> [point-wise posterior covariance] This is one of the limitations of our work. It cannot directly be used to compute a multi-variate predictive distribution over several data points simultaneously. Note, however, that this is also a limitation of the method SWSGP we compare favorably with. Such a method suffers from the same limitation as ours. We  have mentioned now this limitation in the updated manuscript. In any case, most performance and uncertainty quantification metrics do not consider covariances and can be computed without them. Nevertheless, this might be a limitation in applications where covariances are essential. One way to work around this is to consider for prediction the union of input dependent inducing points for test points. However, this would be inconsistent with the training method proposed for IDSGP. Such approach can be modified to take this into account by considering the union of input dependent inducing points for the training points within a mini-batch. This is a solution similar to one proposed in SWSGP for considering covariances.

---

> > ### Comment · Reviewer_oyvg · 2021-11-29
> > **Response to comments and other reviewers**
> >
> > I thank the authors for taking the time to thoughtfully address both my comments and the comments of the other reviewers.  I liked this paper at the start, and continue to like it now.  I hope it will appear in ICLR.

---

### Decision · Program_Chairs · 2022-01-20

**Decision:**

Reject

**Comment:**

Three reviewers recommend Reject. Two reviewers recommend Accept although do not champion the paper. I believe the paper develops an interesting idea to better estimate the location of the inducing inputs in sparse GP models. However, I still think the paper would benefit from another careful revision and therefore I do not recommend Acceptance at this stage. I agree with reviewers that 1) currently the method is unable to estimate the covariance between two data points. This is important in applications of GPs for uncertainty quantification such as Bayesian optimisation. For example, including a BayesOpt example would clearly strengthen the paper. 2) empirical evaluation lacks simple baselines, e.g. Titsias (2009). The authors claim that Titsias (2009) does not scale and that's why they don't care for it. Even if this is true, including an example that helps to better compare against this method at a different scale might strengthen the model proposed here.